# Compensatory ion transport buffers daily protein rhythms to regulate osmotic balance and cellular physiology

Alessandra Stangherlin [1], Joseph L. Watson [1], David C. S. Wong [1], Silvia Barbiero [1], Aiwei Zeng [1], Estere Seinkmane [1], Sew Peak Chew [1], Andrew D. Beale [1], Edward A. Hayter [2], Alina Guna[3], Alison J. Inglis[4], Marrit Putker[1,11], Eline Bartolami[5,12], Stefan Matile[5], Nicolas Lequeux[6], Thomas Pons [6], Jason Day [7], Gerben van Ooijen [8], Rebecca M. Voorhees[4], David A. Bechtold [2], Emmanuel Derivery [1], Rachel S. Edgar[9], Peter Newham [10] & John S. O'Neill [1✉]

Between 6–20% of the cellular proteome is under circadian control and tunes mammalian cell function with daily environmental cycles. For cell viability, and to maintain volume within narrow limits, the daily variation in osmotic potential exerted by changes in the soluble proteome must be counterbalanced. The mechanisms and consequences of this osmotic compensation have not been investigated before. In cultured cells and in tissue we find that compensation involves electroneutral active transport of Na⁺, K⁺, and Cl⁻ through differential activity of SLC12A family cotransporters. In cardiomyocytes ex vivo and in vivo, compensatory ion fluxes confer daily variation in electrical activity. Perturbation of soluble protein abundance has commensurate effects on ion composition and cellular function across the circadian cycle. Thus, circadian regulation of the proteome impacts ion homeostasis with substantial consequences for the physiology of electrically active cells such as cardiomyocytes.

[1] MRC Laboratory of Molecular Biology, Cambridge, UK. [2] Faculty of Biology, Medicine and Health, University of Manchester, Manchester, UK. [3] UCSF, San Francisco, CA, USA. [4] California Institute of Technology, Pasadena, CA, USA. [5] Department of Chemistry, University of Geneva, Geneva, Switzerland. [6] LPEM - ESPCI Paris, PSL, CNRS, Sorbonne Université, Paris, France. [7] Department of Earth Sciences, University of Cambridge, Cambridge, UK. [8] School of Biological Sciences, University of Edinburgh, Edinburgh, UK. [9] Department of Infectious Diseases, Imperial College London, London, UK. [10] Clinical Pharmacology and Safety Sciences, R&D, AstraZeneca, Cambridge, UK. [11] Present address: Crown Bioscience Netherlands B.V., Utrecht, The Netherlands. [12] Present address: CEA, IRIG, SyMMES, Grenoble, France. ✉email: oneillj@mrc-lmb.cam.ac.uk

The abundance of many proteins exhibits ~24 h rhythms, regulated by cell-autonomous circadian timing mechanisms that align physiology with the day–night cycle[1–6]. Between 6 and 20% of cellular proteins are under circadian control, and the expression of most oscillating proteins peaks during translational "rush hours" that typically coincide with the organism's habitual active phase[1–7]. Daily regulation of mechanistic target-of-rapamycin complexes (mTORC) partitions phases of increased protein production (increased mTORC activity) from those of increased catabolism (decreased mTORC activity)[8–11]. This temporal organization of protein homeostasis (proteostasis) permits the most efficient use of bioenergetic resources, both in vivo and in cultured mammalian cells[8–10,12–14]. How cellular protein concentration varies over circadian time in different intracellular compartments is currently unknown.

In the crowded cytosol, macromolecules (300–550 mg/mL[15]) and K⁺ ions (~145 mM[16]) are the major determinants of cytosolic osmotic potential, balanced by high extracellular concentrations of Na⁺ and Cl⁻ ions. Cells must therefore accommodate any daily variation in cytosolic protein abundance without compromising osmotic homeostasis (osmostasis), which would have deleterious effects on cellular function and viability[17–20]. Regulatory volume increase/decrease (RVI/D) describes the several osmoregulatory mechanisms that act in concert to protect cell volume against acute changes in extracellular tonicity[17,21]. However, very little is known about how cells compensate for physiological changes in intracellular macromolecule content to remain in osmotic equilibrium during the circadian cycle.

Major components of RVI/D are the electroneutral cotransporters of the SLC12A family: ubiquitously-expressed symporters that regulate transmembrane ion gradients and cell volume across a wide variety of tissues[22]. These chloride cotransporters couple ion transport with the respective transmembrane Na⁺ and K⁺ gradients that are ultimately established by the universal and essential Na/K-ATPase antiporter[17]. The ubiquitous Na-K-Cl (NKCC1/SLC12A2) cotransporter facilitates electroneutral secondary active ion influx (import of 1 Na⁺, 1 K⁺, with 2 Cl⁻), whereas the ubiquitous K-Cl (KCC1/SLC12A4) cotransporter facilitates electroneutral ion efflux (export of 1 K⁺ and 1 Cl⁻)[22]. KCC3/SLC12A6 and KCC4/SLC12A7 function similarly to KCC1 and are widely expressed, whereas paralogous KCC2 is neuron-specific. NKCC2/SLC12A1 and NCC/SLC12A3 expression are restricted to a few tissues[23,24], while CIP/SLC12A9 and CCC9/SLC12A8 are widely expressed but poorly characterized[25]. Changes in the relative activity of NKCC and KCC isoforms regulate intracellular ion levels, Cl⁻ in particular. During neuronal development, for example, progressively increasing KCC2 activity reduces intracellular Cl⁻ and mediates the transition between excitatory and inhibitory GABA signaling[26,27]. Furthermore, differential KCC/NKCC activity is thought to mediate GABAergic synaptic plasticity of circadian regulation in the mammalian suprachiasmatic nucleus[28,29].

Several SLC12A cotransporters show high amplitude circadian rhythms at the mRNA level[4,30], but their activity is primarily regulated post-translationally via a coordinated network of kinases, including the essential serine/threonine kinase oxidative stress response kinase 1 (OXSR1) and its paralog SPAK[17,31]. A large body of evidence supports the differential regulation of SLC12A cotransporters by OXSR1/SPAK: phosphorylation of NKCC allows ion import (Na⁺, K⁺, and Cl⁻) whilst phosphorylation of KCC attenuates ion export (K⁺ and Cl⁻)[31,32]. The activity of OXSR1/SPAK is regulated upstream through phosphorylation by lysine deficient protein kinase family members (WNKs), which are stimulated by molecular crowding and low cytosolic Cl⁻[31–33]. Daily variation in the phosphorylation of

WNK isoforms has been reported in mouse liver[12] and forebrain[30], as has the phosphorylation of several members of the SLC12A family[30,34]. Importantly, inactivating mutations of genes encoding SLC12A family members, WNK paralogs, and OSR1/SPAK underlie a wide range of inherited human diseases, including pseudohypoaldosteronism type II, Gitelman and Bartter syndromes, highlighting the importance of the WNK-OXSR1/SPAK-SLC12A pathway for normal physiology[31,35–37].

Here, we describe a fundamental and isovolumetric process whereby electroneutral ion transport buffers intracellular osmotic potential against daily and acute variations in cytosolic soluble protein abundance. Consequently, the ionic composition of the cell is not constant throughout the day, as has been assumed[38]. We found that dynamic changes in ion abundance drive oscillations in cellular physiology that impart temporal regulation to cardiomyocyte cell function and heart rate. More broadly, our data suggest that the cellular capacity for dynamic ion transport is important for protein homeostasis.

## Results

**Cell-autonomous rhythms in soluble protein abundance.** Up to a fifth of the cellular proteome exhibits 24 h cycles in abundance[1–7], due in part to circadian oscillations of mTORC1 activity, ribosome biogenesis, and global protein synthesis rates[10,39,40]. We confirmed that mTORC1 activity rhythms occur cell-autonomously in quiescent adult mouse fibroblasts, maintained under constant conditions (37 °C), following synchronization by daily temperature cycles (12 h:12 h 32 °C:37 °C) (Fig. 1a, Supplementary Fig. 1a).

mTORC1 activation stimulates translation[41], cytoplasmic targeting of many different macromolecules, and macromolecular crowding[42]. We therefore asked whether we could detect circadian oscillations in the level of soluble cytosolic protein, since many rhythmically abundant proteins localize to this compartment. Total protein lysates and cytosolic soluble extracts (hereafter "soluble protein") were prepared from fibroblast cultures over three consecutive days, under constant conditions as above (Supplementary Fig. 1a), and selective lysis of the plasma membrane by digitonin was confirmed by mass spectrometry (Supplementary Fig. 1b, c)[43]. While total cellular protein showed no oscillation, we detected clear ~24 h rhythms in the concentration of soluble protein extracted from cells (Fig. 1b, Supplementary Fig. 1d). Peaks were observed 4–6 h after the peak of the clock reporter PER2::LUC (PERIOD 2-firefly LUCIFERASE fusion reporter[44]), coinciding with both maximal mTORC1 activity (Fig.1a) and increased translation rate in the soluble fraction (Fig. 1c).

These findings were validated using cellular fractionation by differential centrifugation, following lysis by NP-40 at biological times corresponding to the peak and trough of the soluble protein rhythm (Fig. 1d). While total protein levels did not change, we observed reciprocal time-of-day variation in protein abundance between the cytosolic and nuclear/organellar fractions. This is compatible with a cell-autonomous daily rhythm in the sequestration of cytosolic proteins to other cellular compartments or membrane-less organelles, which has been observed in several model organisms[45–47]. Whether daily cycles of cytosolic protein levels are primarily attributable to rhythmic protein synthesis/degradation or rhythmic protein sequestration/liberation warrants further investigation. However, we can exclude pH-dependent changes in cytosolic protein solubility, since no variation in cytosolic pH was detected over the cellular circadian cycle (Supplementary Fig. 2).

Macromolecular crowding describes the concentration-dependent effect that overall macromolecular abundance has on

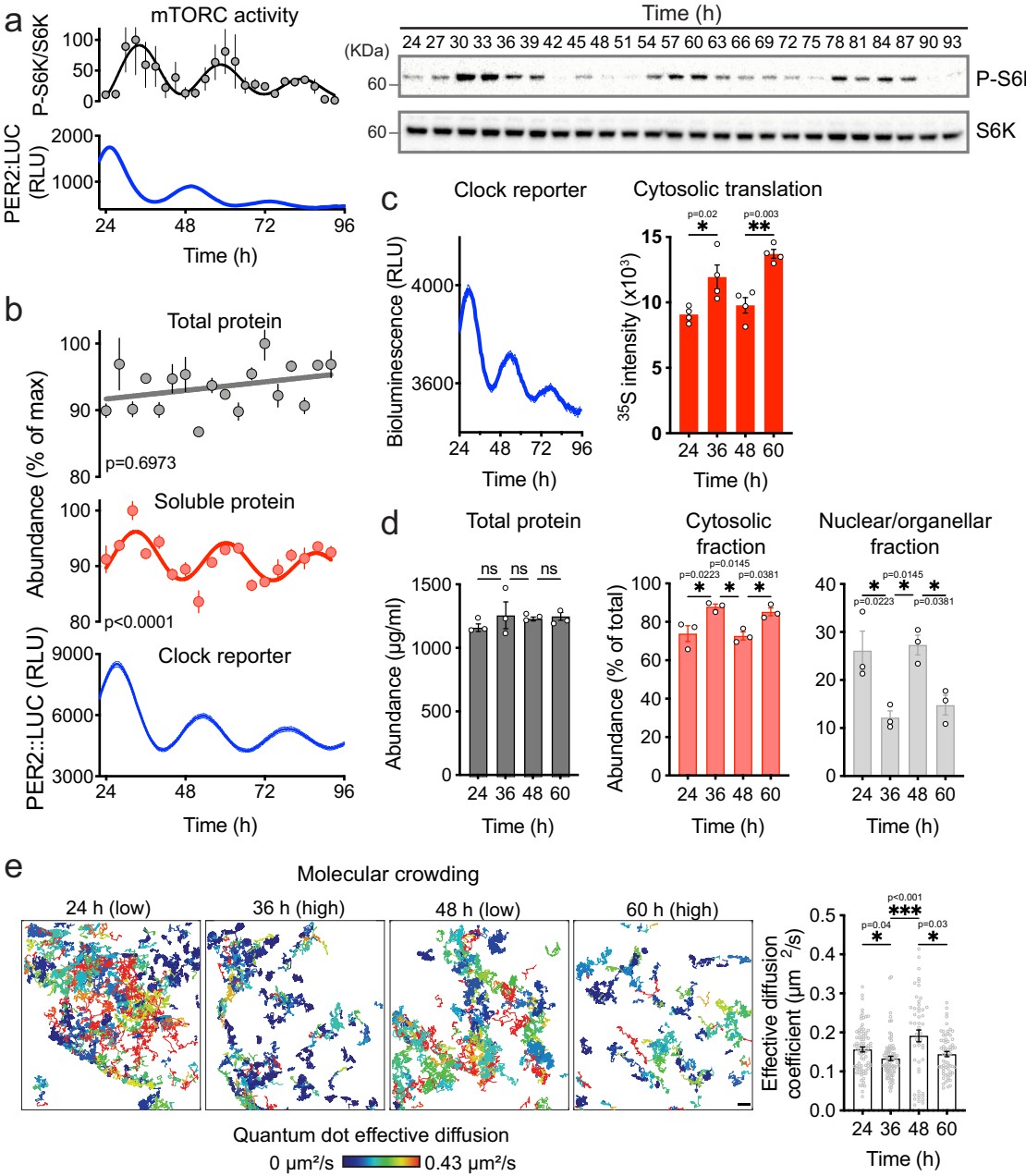

**Fig. 1 Circadian variation in cytosolic protein content in mouse fibroblasts. a** Cell-autonomous circadian mTORC1 activity detected by immunoblots of phospho-S6 kinase and S6 kinase abundance in fibroblasts sampled every 3 h over 3 days in constant conditions ($n = 3$) (right). Blot quantification and parallel PER2::LUC bioluminescent clock reporter ($n = 3$) (left). **b** Total and soluble protein quantification of cell lysates sampled ever 4 h ($n = 3$) and parallel PER2::LUC bioluminescent clock reporter in fibroblasts ($n = 3$). Protein abundance values were normalized for the maximal value in each timeseries. **c** 35S-met/cys incorporation assay at peak (36 h/60 h) and trough (24 h/48 h) of S6K phosphorylation rhythms ($n = 4$) and parallel PER2::LUC activity ($n = 3$). **d** Protein quantification of total cell lysates, nuclear/organellar, and cytosolic fractions at peak (36 h/60 h) and trough (24 h/48 h) of protein rhythms ($n = 3$). **e** Effective diffusion rate of QDs in fibroblasts ($n = 81, 83, 51, 63$, respectively) and representative tracking images. Color key represents diffusion rate. Scale bar: 1 μm. Mean ± SEM shown throughout. p-values in (**b**) indicate comparison of damped cosine wave with straight-line fit (null hypothesis = no rhythm). Significance was calculated using one-way ANOVA and Tukey's multi comparisons test (MCT) (**c**), one-way ANOVA and Sidak's MCT (**d**), and Kruskal-Wallis with Dunn's MCT (**e**). Cell type used was lung fibroblasts for (**a–d**) and cardiac fibroblasts for (**e**).

the activities of macromolecules, as well as on the bulk solvent. Diffusion of macromolecules in the cytosol is sensitive to several factors, particularly macromolecular crowding i.e., as macromolecule concentration increases, the diffusion of macromolecules decreases[48–50]. This can be measured directly from the diffusion of inert quantum dot nanoparticles (QDs, 20 nm diameter), both in solution and in cells (Supplementary Fig. 3 a–c). Proteins are the most abundant macromolecule in

mammalian cells[51]. Therefore, to further validate the observed rhythm in cytosolic protein concentration, we measured the diffusion of QDs in the cytosol[52], reasoning that changes in cytosolic protein concentration during the circadian cycle should inversely impact the diffusion of QDs. In line with prediction, we observed time-of-day variation in the effective diffusion coefficient of QDs, with faster diffusion occurring when protein abundance is low (Fig. 1e, Supplementary Video 1 and 2).

Importantly, mTORC inhibition with a low, subsaturating concentration of the inhibitor torin1[53] attenuated the peak of soluble protein rhythms (Supplementary Fig. 4a), and commensurately diminished the daily variation in molecular crowding reported by QD diffusion (Supplementary Fig. 4b). As would be expected, the effect of torin1 on soluble protein and crowding was most apparent at biological times when mTORC1 is maximally active, compared with 12 h later. Overall, these observations suggest that there are cell-autonomous daily rhythms in cytosolic soluble protein abundance and molecular crowding. These rhythms are at least partially dependent on cell-autonomous circadian regulation of mTORC1 activity, through some combination of rhythmic protein production and differential association of cytosolic proteins with other cellular compartments.

**Antiphasic oscillations in soluble protein and intracellular ion abundance.** Most cells are highly permeable to water, and therefore susceptible to changes in volume upon osmotic challenge. Tissue fluid homeostasis ensures that cells are rarely subject to acute perturbations in extracellular osmolarity, but they must defend cell volume against fluctuations in intracellular osmolarity[6,17]. Indeed, within the densely crowded cytosolic compartment (where [macromolecules] is 300–550 mg/mL[15]) small changes in protein concentration elicit large changes in osmotic potential (Supplementary Fig. 5). Without compensatory mechanisms, the high amplitude rhythm in soluble protein we observe would result in potentially deleterious oscillations of cell volume, as water moves over the plasma membrane to maintain osmotic equilibrium[21]. As we have observed previously, however[2], there was no commensurate daily variation in cell volume (Fig. 2a, Supplementary Fig. 6a), meaning that cells successfully buffer osmotic potential against gradual changes in cytosolic protein concentration over circadian timescales.

In light of prior observations[9,45,54], we hypothesized that transmembrane ion flux was responsible for this isovolumetric compensatory mechanism. We focused on metal ions, the most abundant osmotically active species in cells (Supplementary Table 1), and analysed the intracellular ionic composition of primary fibroblasts across three circadian cycles by inductively coupled plasma-mass spectrometry (ICP-MS)[55]. Importantly, we observed ~24 h ion rhythms for intracellular $K^+$, $Na^+$, and $Cl^-$, which oscillated in antiphase with soluble protein (Fig. 2b, Supplementary Table 2): ions are exported when soluble protein increases and imported when soluble protein decreases. Several other biologically relevant ions were not similarly rhythmic (Supplementary Fig. 6b).

To assess whether the ion and cytosolic protein rhythms are under cell-autonomous circadian control, we analysed cells from short period *tau* mutant mice[56], which express circadian rhythms that run faster than wild type. Soluble protein and ion abundance oscillated in antiphase, as with wild-type cells, but showed correspondingly shorter period oscillations (~21 h) (Fig. 2c, Supplementary Fig. 6c). In a parallel investigation, which compared CRYPTOCHROME-deficient with wild-type fibroblasts[57], we found that the amplitude of soluble protein rhythms correlated directly with the amplitude of cellular $K^+$ rhythms, which remained in antiphase with each other. These data suggest that cell-autonomous circadian clock mechanisms coordinate compensatory ion transport to accommodate daily variation in cytosolic protein content, without volume change.

**Reciprocal regulation of soluble protein and intracellular ion abundance.** In concert with other transport systems, the ubiquitous and essential $Na^+/K^+$-ATPase establishes ~10-fold ion gradients over the plasma membrane, with high $K^+$ and low $Na^+/Cl^-$ in the cytosol (Supplementary Table 1). In response to the acute external osmotic challenge, the net movement of water rapidly re-establishes osmotic equilibrium, with resultant volume changes then being quickly countered by solute transport that returns the cell to its original volume[17,58]. This homeostatic mechanism, known as regulatory volume increase/decrease (RVI/D), is common to all mammalian cells, and its molecular mechanisms are very well characterized[17,22,59–62]. $K^+$ is the most abundant cytosolic osmolyte, and as such $K^+$-transport via SLC12A cotransporters constitutes a major component of RVI/D. All SLC12A family members transport $Cl^-$ as obligate counterions, presumably this is essential for their function since electrostatic constraints would prevent large-scale net cation efflux or import.

RVI/D solute transport systems act to maintain volume and osmotic homeostasis against external challenge. Given the isovolumetric and antiphasic rhythms of ions and soluble protein we observed, it seemed plausible that RVI/D solute transport systems also maintain volume and osmotic homeostasis against internal osmotic challenge, i.e., daily changes in soluble protein concentration are counterbalanced by reciprocal changes in RVI/D activity but without preceding volume change. This would predict that the ability of a cell to accommodate increases in cytosolic protein should be limited by the capacity to effect compensatory ion export via RVI/D mechanisms, of which the SLC12A cotransporters are major components. To test this, we serum-stimulated non-synchronized cells to elicit an acute mTORC1-mediated increase in soluble protein concentration. This soluble protein increase was abolished by NKCC1 knock-down, as well as by transient inhibition of either NKCC1 or KCC1-4: SLC12A members that facilitate the influx of $Na/K^+/2$ $Cl^-$ and efflux of $K^+/Cl^-$, respectively (Supplementary Fig. 7a, b). This indicates that both KCC and NKCC1 activities are permissive for acute increases in soluble protein levels stimulated by serum factors. Indeed, the combined inhibition of both the NKCC and KCC abolished serum-induced protein increases as effectively as inhibition of mTORC itself (Fig. 2d).

Because intracellular ion abundance varies across the day, we predicted that at biological times when ion abundance is already low, the reduced cellular capacity to buffer osmotic potential should constrain any further increase in cytosolic protein concentration. Consistent with this prediction, in synchronized cells no increase in soluble protein was detected upon serum stimulation of cells at the minima of ion rhythms (36 and 60 h, when soluble protein is already high, Fig. 2e). Moreover, serum stimulation elicited no increase in soluble protein at any timepoints in cells under mTORC inhibition. These findings suggest a bidirectional dynamic interplay between the transporter activity and protein levels: whilst mTORC1-dependent changes in cytosolic protein abundance stimulate compensatory ion transport, the capacity for ion transport restricts the scope for acute, as well as daily, increases in protein.

**Daily regulation of SLC12A transporter activity.** In the context of RVI/D, the coordinated activity of SLC12A family members is thought to be regulated via WNK-OXSR1/SPAK1 signaling, stimulated by low $Cl^-$ and increased macromolecular crowding[32,63–65] (Fig. 3a). We tested and validated predictions from the current consensus model with a kinase assay for WNK autophosphorylation in vitro and with cellular assays for OSXR1 activity (Fig. 3b, c, respectively).

Low $Cl^-$ and increased macromolecular crowding occur coincidentally during the circadian cycle, when mTORC1 activity is high (Fig. 2c, Supplementary Fig. 8a), leading to the prediction that OXSR1 signaling and net SLC12A-mediated ion transport should exhibit circadian regulation. Consistent with this, in

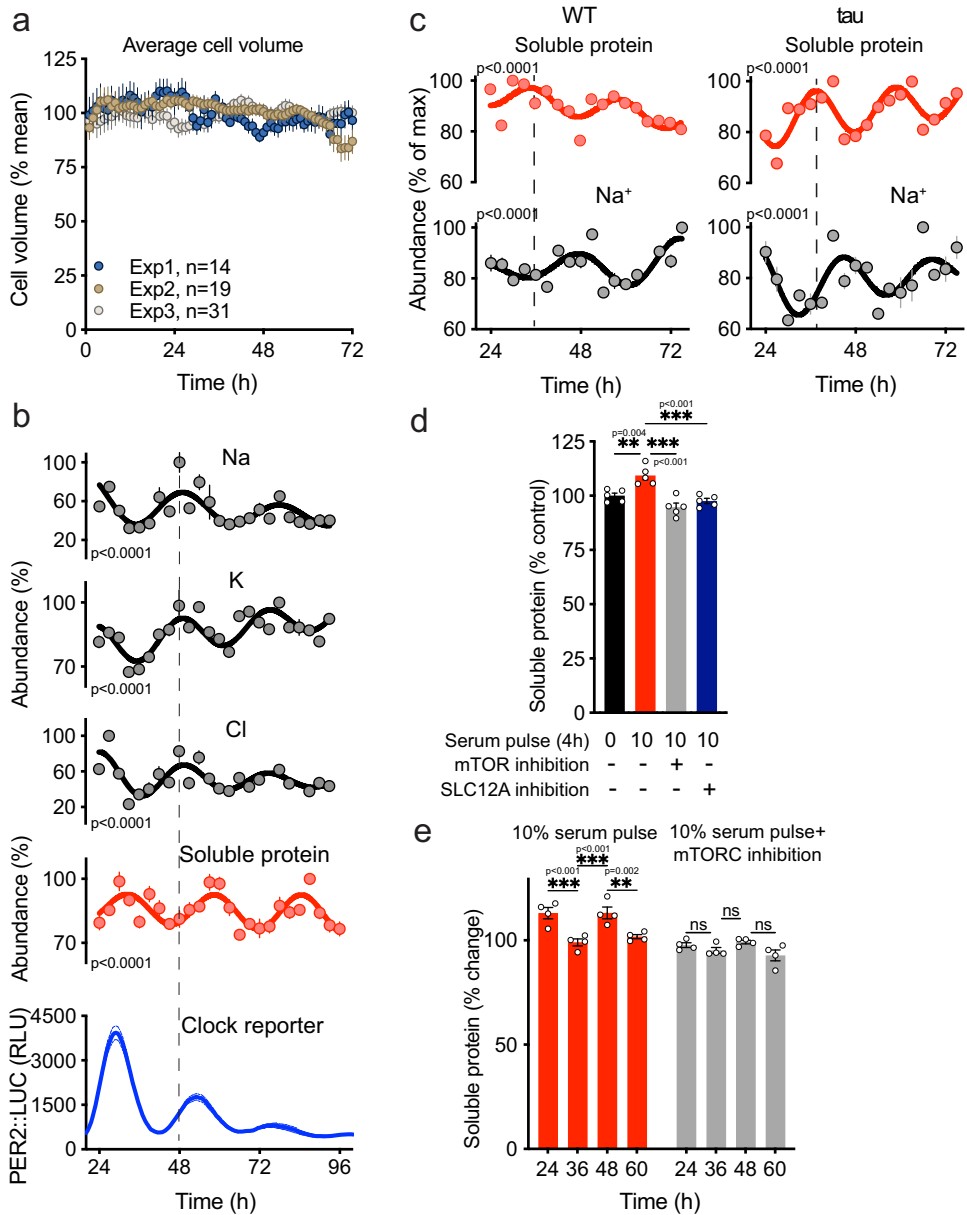

**Fig. 2 Ion flux buffers changes in cytosolic protein abundance in mouse fibroblasts. a** Mean cell volume of fibroblasts across time from three independent recordings. Average cell volume data were normalized for the average of all values within each replicate. **b** Abundance of selected ions ($n = 4$), soluble protein ($n = 3$), and PER2::LUC bioluminescence recordings ($n = 3$) in fibroblasts. Abundance values were normalized for the maximal value in each timeseries. **c** Quantification of intracellular $Na^+$ ($n = 4$) and labile cytosolic protein abundance ($n = 3$) in WT and *tau* mutant fibroblasts under constant conditions over 2 days. **d** % change in soluble protein upon 4 h treatment with 10% serum ± 50 nM mTOR inhibitor torin1 or 10% serum ± 50 μM DIOA and 100 μM Bumetanide ($n = 3$). **e** % fold increase in soluble protein upon 4 h treatment with 10% serum ± 50 nM mTOR inhibitor torin1 at indicated times ($n = 4$). Mean ± SEM shown throughout. *p* values in (**b**) and (**c**) indicate comparison of damped cosine wave with straight-line fit (null hypothesis = no rhythm). Statistical tests were one-way ANOVA and Tukey (**d**) and two-way ANOVA and Sidak's MCT (**e**). Cell type used was lung fibroblasts for (**c**) and cardiac fibroblasts for (**a, b, d, e**).

cultured fibroblasts, the phosphorylation profile of OXSR1 showed cell-autonomous daily rhythms (Fig. 3d), very similar to that of phospho-S6K (Fig. 1a). At no point was no OXSR1 phosphorylation observed, indicating some basal KCC and NKCC activity throughout the circadian cycle. However, peak OXSR1 phosphorylation coincided with the nadir of $Cl^-$ rhythms and maximal macromolecular crowding (Supplementary Fig. 8a). Moreover, several recent phospho-proteomics studies indicate daily variation in the phosphorylation of WNK1-4 paralogs in several tissues, including cultured fibroblasts[57], mouse liver[12] and forebrain[30] (Supplementary Fig. 8b-d), as well as

rhythms in the phosphorylation of NKCC and KCC isoforms[30,34], suggesting that similar mechanisms operate in vivo.

Over the circadian cycle, and in light of established RVI/D mechanisms regulating SLC12A transporter activity, our observations suggest that mTORC-dependent increases in cytosolic protein are initially facilitated by electroneutral active $K^+$-$Cl^-$ export via KCC, which simply functions to maintain osmotic equilibrium over the plasma membrane (Fig. 3a, Supplementary Fig. 9). As cytosolic $Cl^-$ levels fall, they will eventually begin to limit KCC activity, because $Cl^-$ is the essential counterion to $K^+$. However, around this time increased macromolecular crowding

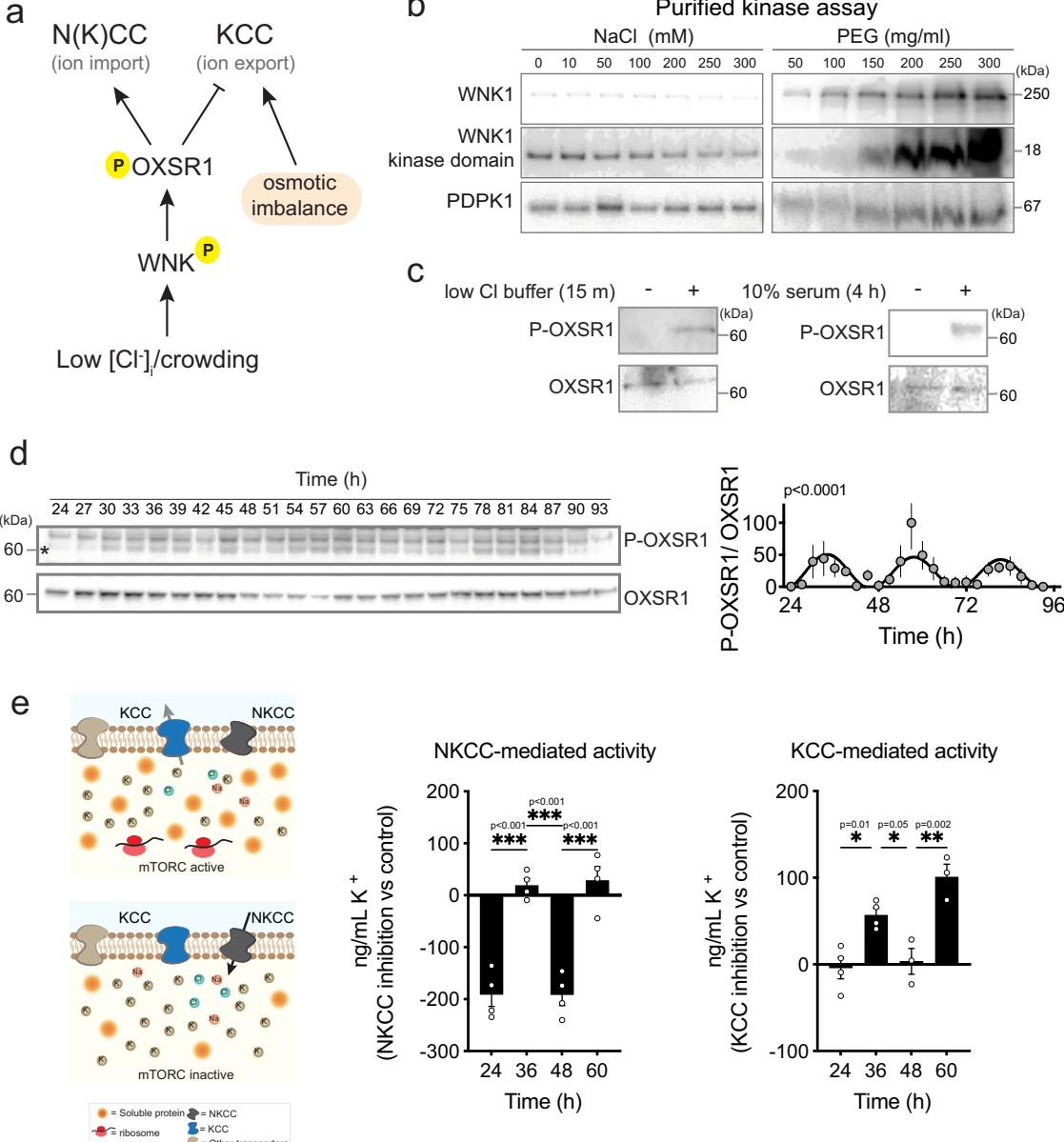

**Fig. 3 Circadian regulation of the WNK/OXSR1/SLC12A pathway activity. a** Schematic of the WNK/OXSR1 pathway and regulation of the N(K)CC and KCC transporters. **b** In vitro kinase activity assays for purified WNK1 and 3-Phosphoinositide Dependent Protein Kinase 1 (PDPK1) upon increasing concentrations of NaCl or polyethylene glycol (PEG) ($n = 1$). WNK1 but not PDPK1 is sensitive to increased macromolecular crowding (mimicked by PEG). Note that WNK1 is inhibited by high concentrations of $Cl^-$. **c** Phosphorylation of cellular OXSR1 upon hypotonic treatment and serum, indicating that decreased intracellular $Cl^-$ and increased cytosolic crowding lead to OXSR1 phosphorylation ($n = 1$). **d** Representative immunoblots and quantification showing phospho-OXSR1 and OXSR1 abundance in fibroblasts sampled every 3 h for 3 days in constant conditions ($n = 3$). The asterisk indicates the P-OXSR1 band. $p$ value indicates comparison of damped cosine wave with straight-line fit (null hypothesis = no rhythm). **e** Model prediction and data showing the difference in cytosolic $K^+$ (digitonin-extracted lysates) between cells treated with DIOA (KCC inhibitor) or bumetanide (NKCC inhibitor) compared with vehicle-treated cells at the indicated time points. Cells were treated for 15 min before sampling ($n = 4$). Statistical test used was two-way ANOVA with Sidak's MCT. Mean ± SEM shown throughout. Cell type used was lung fibroblasts (**c–e**).

acts synergistically with decreased $Cl^-$ levels to stimulate WNK activity[58], which will then stimulate increased import of $Na^+$ and $Cl^-$ via NKCC and other transporters. Cytosolic $Na^+$ is exported by several mechanisms[32,63,64,66], whereas $Cl^-$ influx provides counterions to sustain KCC activity. This allows further increases in cytosolic protein to be accommodated by additional $K^+/Cl^-$ export, facilitated by increased $Cl^-$ flux in both directions. This phase is followed by the circadian decrease in mTORC activity, resulting in gradual reduction of cytosolic protein. The osmotic equilibrium over the plasma membrane is now maintained by

increased import of $Na^+$, $K^+$, and $Cl^-$ via NKCC and other transporters, with KCC now being much less active. Thus, the overall consequence of increased mTORC activity signaling is net $K^+$ export, with much smaller comparative changes in steady-state $Na^+$ and $Cl^-$ levels, whose cellular levels are ~10-fold lower than $K^+$. Whereas, later decreases in soluble protein are balanced by net $K^+$ import.

This model predicts that the selective transient inhibition of KCC or NKCC at the daily peak (36/60 h) or trough (24/48 h) of mTORC activity (Figs. 1a, 3e) will elicit opposite effects on

intracellular $K^+$. We validated these predictions via 15-minute acute inhibition, observing changes in $K^+$ abundance that suggest there are distinct phases of net $K^+$ import and export that render cells more sensitive to KCC inhibition when soluble protein is increasing, and more sensitive to NKCC inhibition when soluble protein is decreasing. Similarly, we also found that sustained OXSR1/SPAK1 inhibition abolished the daily difference in soluble protein levels (Supplementary Fig. 3c), consistent with our model where SLC12A-mediated ion transport is required for return of soluble protein concentration to basal levels, not their increase. The model we propose for N(K)CC-mediated compensatory ion transport pivots around dynamic daily changes in WNK-OSXR1/SPAK1 activity and differential KCC vs. NKCC activity, and is primarily informed by acute inhibition of signaling and transporter function within a circadian cycle, rather than across it. The molecular mechanisms of SLC12A regulation are already well-characterized in the context of RVI/D, following extracellular osmotic challenge[17,20,62,67–69], we simply propose that RVI/D regulation of NKCC1 vs. KCC extends to encompass intracellular osmotic challenge over the circadian cycle without significant volume change.

The SLC12A family includes nine members (including KCC1-4) whose overlapping and semi-redundant functions are important for physiological function and overall viability[37,70,71]. The same is true for the four WNK paralogs, as well as OSXR1 with SPAK1[31,69]. This combination of overall essentiality coupled with the functional redundancy between components, presents a barrier to the use of genetic approaches for delineating the specific contribution of individual transporters and kinases over the circadian cycle. NKCC1 deletion is tolerated, however, apparently through upregulation of other (non-SLC12A) transporter activities[72]. Consistent with previous reports[70,72], NKCC1$^{-/-}$ mice showed obvious growth and developmental defects (small size, deaf, unsteady gait) and weaned in non-Mendelian ratios. Fibroblasts derived from adult PER2::LUC NKCC1$^{-/-}$ mice had significantly reduced cellular $Na^+$ and $K^+$ levels with commensurately increased soluble and total protein levels (Supplementary Fig. 10a). Whilst NKCC1 deletion clearly perturbs steady-state osmostasis[31,37], rather than its daily dynamics, this genetic evidence further validates the reciprocal relationship between cellular ion and macromolecular content, and indicates that NKCC1$^{-/-}$ cells adapt to the loss of this transporter through a permanently altered set point for osmotic homeostasis. Importantly, NKCC1$^{-/-}$ cells showed a similarly proportional increase in soluble protein following serum stimulation with respect to wild type, which our model predicts to be initially KCC-mediated, and demonstrates that NKCC1$^{-/-}$ cells remain competent to facilitate an increase in soluble protein levels from an altered baseline (Supplementary Fig. 10b). Critically, however, NKCC1$^{-/-}$ cells showed blunted mTORC1 and OXSR1 activation following serum-stimulation (Supplementary Fig. 10c), consistent with feedback from osmostasis to regulate the means of macromolecular synthesis, and consistent with previous observations[45,72]. Interestingly, NKCC1$^{-/-}$ knockout cells and mice also exhibited lengthened period of circadian rhythms in cellular gene expression and mouse locomotor activity (Supplementary Fig. 11). This is not especially surprising, since mTORC1 signaling regulates circadian rhythms[8], and mTORC1 activation is regulated by the capacity for effecting compensatory ion efflux, which will be less efficient without NKCC1 to facilitate $Cl^-$ recycling according to our model (Supplementary Fig. 9).

**Cell-autonomous rhythms in cardiomyocyte activity and heart rate.** We next considered the physiological consequences of daily rhythms in intracellular ion content. The concentrations of $Na^+$,

$K^+$, and $Cl^-$ across the plasma membrane and their membrane permeability determine cellular resting potential and the electrophysiological properties of excitable cells[38]. We therefore employed primary mouse neonatal cardiomyocytes as a model system in which to test how cell-autonomous circadian regulation of ion abundance might impact upon cellular electrical activity. As with primary fibroblasts, we observed ~24 h cell-autonomous rhythms in intracellular $Na^+$, $K^+$, and $Cl^-$ content (Fig. 4a, Supplementary Fig. 12a). To assess the relevance of such rhythms in vivo, we harvested whole heart tissue every 4 h from adult mice under diurnal conditions and analysed intracellular ion abundance by ICP-MS (Fig. 4b). We observed a significant variation in the intracellular abundance of $Na^+$ and $K^+$, with ion content peaking at biological times equivalent to the end of the rest phase when mTORC activity and soluble protein are normally low[10,11].

Pacemaker cardiomyocytes spontaneously depolarize and fire action potentials. Spontaneous depolarization is due to a mixed $Na^+$ and $K^+$ inward current (Funny-like current)[73,74], which is affected by $K^+$ and $Na^+$ gradients across the plasma membrane. The slope of this slow depolarization phase, known as the pacemaker potential, determines the frequency of action potential firing. Our data predict that depolarization should occur more quickly when intracellular ion concentrations are high, permitting firing to occur with higher frequency. To test this, we cultured primary cardiomyocytes on multielectrode arrays, and measured firing rate over two days, 12 h apart, at biological times corresponding to the peak and trough of ion rhythms. We observed a time-of-day variation in firing rate, with more frequent action potential firing when ion levels are high (Fig. 4c, Supplementary Fig. 12b). To test the relevance of reciprocal regulation of protein and ion abundance, we performed the same experiment upon inhibition of mTORC activity. We reasoned that the attenuation of protein rhythms should flatten ion rhythms and associated oscillations in cellular function. Indeed, torin1 reversibly abolished the time-of-day dependent variation in cardiomyocyte activity (Fig. 4d). Of note, the effect of torin1 on action potential firing frequency was most evident at 12 and 36 h, when mTORC activity and soluble protein are normally high[11]. Taken together, these data are consistent with a model whereby cell-autonomous, time-of-day variation in cardiomyocyte electrical activity is driven, at least in part, by the coupling of cellular ion content to ~24 h rhythms in cytosolic protein abundance.

In line with this model, we observed time-of-day dependent variation in heart rate (HR) in Langendorff-perfused hearts from adult mice. Importantly, the time-of-day tissue-intrinsic variations in HR were abolished by mTORC1 inhibition, consistent with our findings in isolated cardiomyocytes (Fig. 4e, Supplementary Fig. 12c). Diurnal variation in HR is known to be modulated by autonomic nervous system input[75,76], but our data in isolated cardiomyocytes and ex vivo hearts reveal circadian regulation by cell-autonomous mechanisms intrinsic to the heart. Concordant with this, time-of-day variation in HR was observed in vivo under complete autonomic blockade (via intraperitoneal injection of metoprolol and atropine), indicating significant modulation of HR by a circadian clock within the heart in vivo (Fig. 4e, Supplementary Fig. 12d). More specifically, in a parallel study, we found that time-of-day variations in RR and QT intervals were maintained under autonomic blockade, implying local circadian control[77]. Under these conditions, the daily variation in HR was also abolished by mTORC1 inhibition (Fig. 4e, f, Supplementary Fig. 12d). This result suggests that cell-autonomous mTORC1-dependent changes in cardiomyocyte ion content contribute to intrinsic HR, allowing the heart to beat faster around subjective dusk—the biological time when the greatest cardiac output is required (Supplementary Fig. 12e).

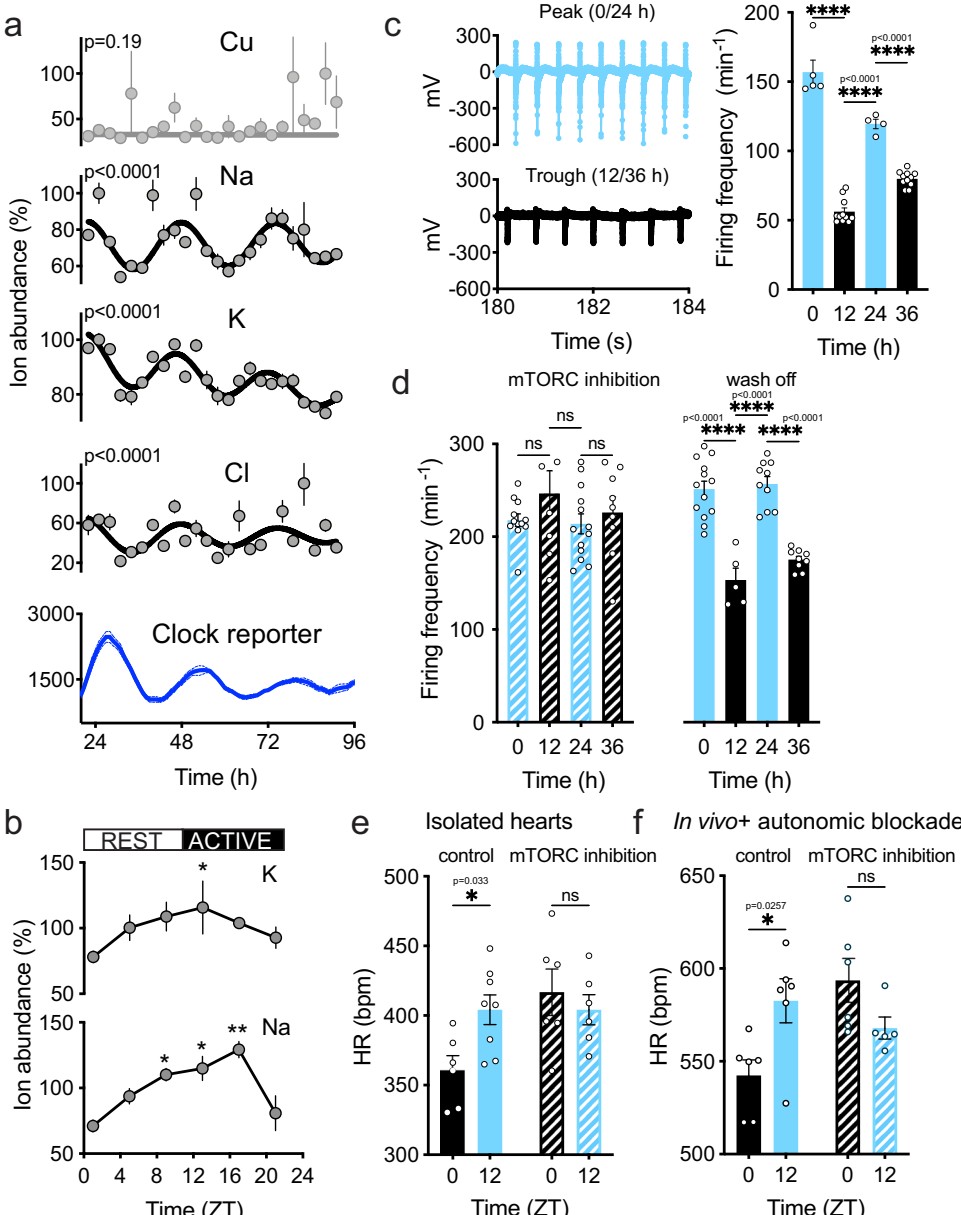

**Fig. 4 Ion fluxes facilitate circadian modulation of cardiomyocyte electrophysiology. a** Abundance of selected ions and PER2::LUC reporter activity in primary cardiomyocytes ($n = 4$) under constant conditions. Values were normalized to maximal values in each timeseries. $p$ values indicate comparison of damped cosine wave with straight-line fit (null hypothesis = no rhythm). **b** Cellular ion content of adult mouse heart tissue under diurnal conditions ($n = 3$, normalized to total protein). **c** Representative field potential traces of cardiomyocytes at peak or trough of ion rhythms and action potential frequency (representative biological replicate, mean signal from active electrodes is shown, $n = 5$, 11, 4, and 11, respectively). **d** Action potential frequency in primary cardiomyocytes at peak and trough of ion rhythms in the presence of the mTOR inhibitor torin1 (50 nM), and wash off from a representative biological replicate (mean values from active electrodes are presented, $n = 12$, 8, 13, and 9 for torin1 and 10, 3, 3, and 5 for wash off). **e** Heart rate (HR) measured ex vivo in Langendorff-perfused hearts from control and rapamycin-treated mice collected at ZT0 and ZT12 ($n = 6$ control ZT0, $n = 8$ control ZT12, $n = 6$ mTORC inhibition ZT0, $n = 8$ mTORC inhibition ZT12). **f** HR measured in vivo by telemetry in control ($n = 6$) or rapamycin-treated mice ($n = 5$) treated with metoprolol and atropine. Time-of-day variation in heart rate persists under complete autonomic blockade. Mean ± SEM shown throughout. Statistical tests are one-way ANOVA with Tukey's MCT (**b**) and (**c**), Two-way ANOVA with Dunnett's MCT (**d**), one-way ANOVA, and Sidak MCT (**e**), and mixed-effect analysis and Sidak MCT (**f**).

## Discussion

Maintenance of osmotic homeostasis and cell volume is a prerequisite for cell viability[19,20], explaining the complex osmoregulatory system that protects cells from acute osmotic insults[17,31]. Very little consideration has been given to the osmotic challenge cells face over circadian time due to intracellular variation in macromolecular concentration, however, or the mechanisms that buffer such daily oscillations and enable the cytosol to remain in osmotic equilibrium with the extracellular environment. We addressed this aspect of cell physiology in mouse fibroblasts, where we identified cell-autonomous rhythms in soluble protein abundance and molecular crowding sustained by daily rhythms of mTORC activity, in the absence of any corresponding alteration in cell volume. By analysing the ionic

composition of the cells across time, we detected antiphasic oscillations in the intracellular abundance of several ions, including $Na^+$, $K^+$, and $Cl^-$: ions are exported upon increases in soluble protein and imported upon decreases in soluble protein. Our data strongly support a model that is entirely consistent with the current understanding of RVI/D, whereby WNK-OXSR1/ SPAK signaling coordinates circadian regulation of ion influx and efflux to facilitate the daily rhythms in net $K^+$ transport through differential regulation of SLC12A transporter activity.

SLC12A members are not the only transporters that facilitate RVI/D under acute challenge[17,78], and so are unlikely to be the only effectors of compensatory circadian solute transport. Nor is $K^+$ the only cellular osmolyte of relevance to osmotic compensation, it is merely the most abundant (by an order of magnitude). Moreover, yeast cells, which lack close homologues of SLC12A family transporters in the plasma membrane, but do have OSXR1/SPAK orthologs, show the same antiphasic rhythms of $K^+$ and soluble protein that we observed in mouse fibroblasts[45]. Thus, whilst we do not discount that other transporters, such as LRRC8 family members[78], contribute to circadian ion fluxes, our observations indicate that SLC12A-mediated electroneutral ion transport is an important mitigator of the osmotic challenge presented by daily rhythms in soluble protein abundance; thereby allowing the cytosolic composition to change each day whilst defending osmostasis and cell volume. Because the transport of small osmolytes buffers intracellular osmotic potential against changes in cytosolic macromolecule concentration, the cellular capacity for $K^+$-export likely confers daily variation upon the capacity of cells to respond to exogenous stimuli through changes in protein expression.

The fundamental rhythmic process we identified has important implications for the physiology of electrically active cells, such as cardiomyocytes, which exhibit the same rhythms in cellular ion content. By performing field potential recordings at biological times corresponding to the peak and trough of ion rhythms, we indeed found a time-of-day variation in the action potential firing rate of isolated cardiomyocytes. Since mTORC inhibition abolished daily firing variation in cardiomyocytes, isolated hearts, and in vivo, mTORC-dependent daily variation in ion content may be the major cell-autonomous factor that imparts circadian organization to cardiac electrical activity. This highlights that soluble protein concentration can directly impact intracellular ion content, and thereby modulate cell electrophysiology.

Diurnal variation in heart rate is largely thought to be regulated by the sympathetic and parasympathetic nervous systems[79]. However, data presented here and our recent study[77] demonstrate that time-of-day variation in heart rate occurs even under complete autonomic blockade, revealing some level of cell-autonomous regulation. Both in vivo and in vitro, the cellular increase in ion content, which increases cell-intrinsic electrical activity, coincides with the beginning of the active phase when basal cardiac output must undergo the greatest shift. We suggest that cell-autonomous cardiomyocyte clocks organize the daily timing of mTORC activity and global protein synthesis/degradation so that the highest cytosolic ion levels occur in anticipation of the biological time-of-day when the heart needs to work hardest (Supplementary Fig. 6e), in concert with central autonomic control. Reduced amplitude of ion rhythms, or their temporal misalignment with sympathetic stimulation, will attenuate the intrinsic capacity of cardiomyocytes to facilitate increased heart rate at this time. For humans, we propose that dysregulation of cardiac ion or protein rhythms, during shift work or aging, renders the heart less competent to satisfy increased output demand in the early morning, contributing to the increased frequency of adverse events at this time[80–85].

To arrive at this new understanding for the temporal regulation of cellular physiology, we focused primarily on cultured fibroblast cells, a most tractable model of the cellular mammalian clock. However, the same timekeeping mechanisms, rhythms in mTORC activity, and daily variation in ion transport have been observed in other cellular contexts, both ex vivo and in vivo[8–11,29,86]. Based on our similar recent findings in human, algal, and fungal cells[9,45], we predict that rhythms in ion transport will be observed in any eukaryotic cell with oscillations in mTORC activity, circadian, or otherwise. Future work is needed to explore the many potential cellular and physiological consequences of ion transport rhythms for other aspects of cellular function, as well as identifying which proteins change their cytosolic localization over the course the circadian cycle and how this impacts their activity.

Finally, our data suggest that reciprocal regulation of ion and protein abundance is a ubiquitous cellular mechanism for osmotic homeostasis, which we expect will be of broad relevance to understanding human physiology and disease. For example, since the capacity to buffer cytosolic osmotic potential is reduced when cytosolic ion levels are low, it is tempting to speculate that this may render neural cells more susceptible to protein misfolding and aggregation toward the end of the daily activity cycle, when most mammals normally seek to rest and sleep.

## Methods

**Fibroblast isolation, culture, and entrainment.** Lung fibroblasts were isolated from adult lung tissues[87] of WT (C57BL6), WT PER2::LUC, and TAU PER2::LUC[56] mice and immortalized by serial passage. Fibroblasts were cultured in DMEM high glucose (Thermo Fisher Scientific, 31966-021) supplemented with 10% serum (GE Healthcare, Hyclone™III SH30109.03) and Penicillin/Streptomycin (Pen/Strep) (Thermo Fisher), at 5% $CO_2$ and 37 °C.

Primary cardiac fibroblasts and cardiomyocytes were isolated from the hearts of P2-P3 neonatal mice using a commercially available dissociation kit following manufacturer's instructions (Miltenyi Biotec, 130-098-373). In brief, heart tissue was removed and stored on ice-cold dissociation buffer (106 mM NaCl, 20 mM HEPES, 0.8 mM $NaH_2PO_4$, 5.3 mM KCl, 0.4 mM $MgS0_4$, 5 mM glucose). Blood vessels and connective tissue were removed and the remaining cardiac tissue was dissected into small sections and digested. After enzymatic dissociation and enzyme inactivation, the digested tissue was filtered through a 70 μm filter (Greiner bio-one, 542070) and placed in 10 cm dishes for 2 h. The supernatant containing cardiomyocytes was collected and seeded in 96 well plates or microelectrode array chips at 1000 cells/mL density using "first day medium" (DMEM high glucose supplemented with 17% M199 (Thermo Fisher, 31150-055), 10% horse serum, 5% newborn calf serum, Glutamax, Pen/Strep, and Mycozap (Lonza)). On the day after the isolation, the medium was replaced with "second day medium" (DMEM high glucose supplemented with 17% M199, 5% horse serum, 0.5% newborn calf serum, Glutamax, Pen/Strep, and Mycozap). The preplating step allows for positive selection of primary cardiac fibroblasts that were then either kept primary (5% $O_2$ incubator) or immortalized by serial passage.

The type of fibroblast used (either lung or cardiac) is indicated in the figure legends. Based on previous experiments we believe that there is no difference in circadian phenotype between the fibroblasts of different tissue origin.

**Bioluminescence recordings.** Cells were seeded and grown to confluence. Cells were then synchronized using temperature cycles of 12 h at 32 °C followed by 12 h at 37 °C for a minimum of 3 days. Bioluminescence was recorded using a culture medium supplemented with 1 mM luciferin (and drugs where indicated) at 5% $CO_2$. Cells were imaged using ALLIGATORs (Cairn Research) as described previously[88], where bioluminescence was acquired for 29 min at 30-min intervals, at constant 37 °C. Mean pixel intensity from each region of interest was extracted using Fiji[89]. For period calculation, bioluminescence traces were detrended by 24 h moving average and fit using GraphPad Prism with a damped cosine wave (where $y$ is the signal, $x$ the corresponding time, the amplitude is the height of the peak of the waveform above the trend line, $k$ is the decay constant (such that $1/k$ is the half-life), phase is the shift relative to a cos wave and the period is the time taken for a complete cycle to occur). Cellular bioluminescence measurements in Supplementary Fig. 2 employed a MOPS-buffering system in place of bicarbonate to allow pH to be adjusted (DMEM powder (Sigma D5030), 0.35 mg/mL sodium bicarbonate, 5 mg/mL glucose, 0.02 M MOPS, 100 U/mL penicillin, 100 μg/mL streptomycin), as described previously[88]. Supplementary Fig. 2c–f employed a Tecan Spark 10 M, whereas Supplementary Fig. 2g employed a Lumicycle (Actimetrics).

**Protein extraction and quantification**. For cytosolic soluble protein extraction, cells (equal number per replicate and per condition) were washed twice with phosphate-buffered saline (PBS) and incubated with a digitonin-based lysis buffer (50 mM tris pH 7.4, 0.01% digitonin, 5 mM EDTA, 150 mM NaCl, protease, and phosphatase inhibitors (Roche, 4906845001 and 04693159001)) on ice for 15 mins. Only supernatant was collected (without scraping). For total protein determination, cells were washed twice with PBS and incubated with RIPA buffer (150 mM NaCl, 1% NP-40, 0.5% Na deoxycholate, 0.1% SDS, 50 mM Tris pH7.4, 5 mM EDTA, protease, and phosphatase inhibitors) on ice for 15 min. After scraping, lysates were collected and sonicated using a Bioruptor sonicator (Diagenode) (30 s ON, 30 s OFF). Protein quantification was performed using intrinsic tryptophan fluorescence[90] or BCA assay (Pierce, 23227). Reported in the figures are protein concentration values normalized as indicated in the axes or in the figure legend.

**Subcellular fractionation**. Wild-type PER2::LUC lung fibroblasts were seeded in 100 mm dishes (300,000 cells per dish) and grown until confluent in temperature cycles (12 h 37 °C–12 h 32 °C) for 7 days. Twenty-four hours before the experiment, cells received a last medium change and were moved into constant conditions. Cytosolic, organelle, and nuclear fractions were harvested every 12 h for 2 days using the Nuclei EZ Prep nuclei isolation kit (Sigma NUC-101) according to manufacturer's instructions. Briefly, cells were washed with ice-cold PBS, lysed with Nuclei lysis buffer, scraped, and centrifuged at 500 g at 4 °C for 5 min. Nuclei pellets were flash frozen. Supernatants were further clarified at 21,000 g at 4 °C for 30 min to get cytosolic fractions, pellets were saved as the organelle fraction, and both fractions were flash frozen. Once thawed, nuclei and organelle pellets were resuspended in 200 μL RIPA buffer (150 mM NaCl, 5 mM EDTA pH 8.0, 50 mM Tris pH 8.0, 1% NP40, 0.5% sodium deoxycholate, 0.1% SDS, protease, and phosphatase inhibitors), sonicated 4 times at high power for 30 s at 4 °C to shear genomic DNA, and centrifuged at 21,000 g for 15 min at 4 °C. Protein concentration was measured using BCA assay (Pierce).

**Quantification of cholesterol in cell fractions**. Wild-type PER2::LUC lung fibroblasts were seeded in 100 mm dishes and grown until confluent in temperature cycles (12 h 37 °C–12 h 32 °C) for 7 days. Twenty-four hours before the experiment, cells received a last medium change and were moved into constant conditions. Cells were fractionated into nuclear, mitochondrial, membrane, and cytosolic fractions every 12 h for 2 days based on a modified previously described protocol[91]. At each timepoint, cells were washed with ice-cold PBS, scraped into 1.5 mL tubes, and lysed with 500 μL fractionation buffer (20 mM HEPES pH 7.4, 10 mM KCl, 2 mM MgCl₂, 1 mM EDTA, 1 mM EGTA, 1 mM DTT, protease, and phosphatase inhibitors) for 15 min on ice. Cells were lysed by syringe extrusion (10 times through a 27-gauge needle) and flash frozen. After thawing, the lysate was centrifuged at 720× g for 5 min at 4 °C and the supernatant was transferred to a fresh tube. The pellet (nuclear fraction) was resuspended in 500 μL RIPA buffer and passed through a 25-gauge needle. The supernatant was further centrifuged at 10,000× g for 5 min at 4 °C and the supernatant was transferred to 0.5 mL ultracentrifuge tubes (Beckman Coulter). The pellet (mitochondrial fraction) was resuspended in 200 μL RIPA buffer and passed through a 25-gauge needle. The supernatant was ultracentrifuged at 100,000× g for 1 h at 4 °C in a TLA120.1 rotor (Beckman Coulter). The supernatant (cytoplasmic fraction) was removed and the pellet (membrane fraction) was resuspended in 200 μL RIPA buffer and passed through a 25-gauge needle. In addition, at each timepoint a parallel dish of cells representing whole cell lysate was lysed in 500 μL RIPA buffer and flash frozen. Each fraction and the whole cell lysate was sonicated 4 times on higher power for 30 s and centrifuged at 21,000× g for 15 min at 4 °C.

Cholesterol content in each sample was quantified using the Amplex Red Cholesterol Assay (Invitrogen, A12216) according to manufacturer's instructions. Briefly, a standard curve was prepared using the provided cholesterol reference standards. Twenty-five microliter of each standard and sample was diluted 2x in 1x Reaction Buffer, added in triplicate to 96 well plates and 50 μL of Amplex Red Reagent working solution was added (containing 300 μM Amplex Red Reagent, 2 units/mL horseradish peroxidase, 2 units/mL cholesterol oxidase, and 0.2 units/mL cholesterol esterase). The plates were incubated at 37 °C for 30 min and a Tecan Spark plate reader was used to measure absorption at 550 nm. The cholesterol content of the cytoplasmic fraction was below the detection limit.

**NKCC1 knockdown**. Lung fibroblasts were seeded in 12 well plates and transfected with either 5 μM of ON-TARGETplus Mouse Slc12a2 siRNA (L-044448-01-0005, Horixon Discovery) or ON-TARGETplus Nontargeting Control Pool (D-001810-10-05, Horizon Discovery) using DharmaFECT 1 Transfection Reagent 1 (T-2001-02, Horizon Discovery) following manufacturer's instructions. After 72 h, cells were moved to serum-free DMEM (Thermo Fisher Scientific, 31966-021) for 24 h and then stimulated with DMEM supplemented with 10% Hyclone III for 4 h. For knockdown validation total cell lysates were extracted with RIPA buffer, whereas soluble protein extracts were extracted using a digitonin-based lysis buffer (as described above).

**Western blotting**. Equal amounts of total cell extracts were run under denaturing conditions using NuPAGE Novex 4–12% Bis-Tris gradient gels (Life Technologies).

Proteins were transferred onto nitrocellulose membranes using the iBlot system (Life Technologies), with a standard (P0, 8 min) protocol. Nitrocellulose was blocked for 60 min in 5% w/w non-fat dried milk (Marvel) in Tris buffered saline/0.05% Tween-20 (TBST). Primary antibodies used were: Anti-OXSR1 (Abcam, ab224248), 1/1000; anti- Phospho-OXSR1 (Abcam, ab138655), 1/1000; anti-S6K (Cell signaling, 2708), 1/2000; anti-P-S6K (Cell signaling, 9205), 1/1000; anti-NKCC1 (Cell Signaling, 14581), 1/500; anti-GAPDH (Sigma, G9545), 1/2000; HRP-conjugated secondary antibody was antirabbit (Sigma-Aldrich, A6154), 1/5000. Chemiluminescence detection was performed using Immobilon reagent (Millipore) and the Gel-Doc™ XR⁺ system (Bio-Rad). Quantification was performed using the ImageJ gel quantification plugin or Image Lab software (Biorad). To better identify phopho-OXSR1 band, one of the replicate samples was treated with Calf intestinal alkaline phosphatase (Promega, M182A) for 60 min at 37 °C.

**Mass spectrometry and gene ontology analysis**

*Sample preparation.* Unsynchronized cells were washed twice in ice-cold PBS and then lysed at room temperature with a digitonin-based lysis buffer (50 mM tris pH 7.4, 0.01% digitonin, 5 mM EDTA, 150 mM NaCl, protease, and phosphatase inhibitors (Roche, 4906845001 and 04693159001)) on ice for 15 min. The supernatant was collected (without scraping) and processed for mass spectrometry analysis.

*Enzymatic digestion.* Samples were reduced with 5 mM DTT at 56 °C for 30 min and then alkylated with 10 mM iodoacetamide in the dark at room temperature for 30 min. Digestion was performed using mass spectrometry grade Lys-C (Promega) at a protein:Lys-C ratio of 100:1 (w/w) for 4 h at 37 °C followed by trypsin (Promega) digestion at a ratio of 50:1 (w/w) overnight. Digestion was quenched by the addition of formic acid (FA) to a final concentration of 0.5%. Any precipitates were removed by centrifugation at 18,500 g for 7 min. The supernatants were desalted using homemade C18 stage tips containing 3 M Empore extraction disks (Sigma-Aldrich) and 2.5 mg of Poros R3 resin (Applied Biosystems). Bound peptides were eluted with 30–80% acetonitrile (MeCN) in 0.5% FA and lyophilized.

*Tandem mass tag (TMT) labeling.* Dried peptide mixtures from each condition were resuspended in 60 μl of 7% MeCN and 1 M triethyl ammonium bicarbonate was added to a final concentration of 200 mM. 0.8 mg of TMT10plex reagents (Thermo Fisher Scientific) was re-constituted in 41 μl anhydrous MeCN. 30 μl of TMT reagent was added to each peptide mixture and incubated for 1 h at RT. The labeling reactions were terminated by incubation with 7.5 μl of 5% hydroxylamine for 30 min. The labeled samples were pooled into one single sample and speed Vac to remove acetonitrile. Labeled sample was desalted and then fractionated with homemade C18 stage tip using 10 mM ammonium bicarbonate and acetonitrile gradients. Eluted fractions were combined into four fractions, acidified, partially dried down in speed vac and ready for LC-MSMS.

*LC MS/MS.* The fractionated peptides were analysed by LC-MS/MS using a fully automated Ultimate 3000 RSLC nano System (Thermo Fisher Scientific) fitted with a 100 μm × 2 cm PepMap100 C18 nano trap column and a 75 μm × 25 cm, nanoEase C18 T3 column (Waters). Samples were separated using a binary gradient consisting of buffer A (2% MeCN, 0.1% formic acid) and buffer B (80% MeCN, 0.1% formic acid), and eluted at 300 nL/min with an acetonitrile gradient. The outlet of the nano column was directly interfaced via a nanospray ion source to a Q Exactive Plus mass spectrometer (Thermo Scientific). The mass spectrometer was operated in standard data-dependent mode, performing a MS full-scan in the m/z range of 380–1600, with a resolution of 70,000. This was followed by MS2 acquisitions of the 15 most intense ions with a resolution of 35,000 and NCE of 33%. MS target values of 3e⁶ and MS2 target values of 1e⁵ were used. The isolation window of precursor ion was set at 0.7 Da and sequenced peptides were excluded for 40 s.

*Spectral processing and peptide and protein identification.* The acquired raw files from LC-MS/MS were processed using MaxQuant (Cox and Mann) with the integrated Andromeda search engine (v.1.6.6.0). MS/MS spectra were quantified with reporter ion MS2 from TMT 10plex experiments and searched against Mus musculus, UniProt Fasta database (March19). Carbamidomethylation of cysteines was set as fixed modification, while methionine oxidation, N-terminal acetylation were set as variable modifications. Protein quantification requirements were set at one unique and razor peptide. In the identification tap, second peptides and match between runs were not selected. Other parameters in MaxQuant were set to default values.

*Gene ontology enrichment analysis.* Gene ontology enrichment analysis was performed using PANTHER 14.1 (Protein ANalysis THrough Evolutionary Relationships) Classification System[92]. The analysis performed was a statistical enrichment test, using the Mann-Whitney U test and Benjamini-Hochberg correction for multiple comparisons.

*35S incorporation assays*. PER2::LUC lung fibroblasts were differentially entrained for a week with external temperature cycles in serum-free DMEM. At the end of the last cycle, cells were moved into constant conditions. A parallel plate was used for bioluminescence recording of the clock reporter (phase marker). At the indicated times, the cells were pulsed with 0.1 mCi/ml $^{35}$S-L-methionine/$^{35}$S-L-cysteine mix (EasyTag™ EXPRESS35S Protein Labeling Mix, Perkin Elmer) in serum-free, Cysteine/Methionine-free DMEM for 15 min at 37 °C. Afterwards, cells were washed with ice-cold PBS and lysed in digitonin-based buffer (with protease inhibitor tablet, added freshly) on ice. Equal amounts of lysates were run on 4–12% Bis-Tris SDS-PAGE using MES buffer. Gels were then dried at 80 °C and exposed overnight to a phosphorimager screen. Images were acquired with Typhoon FLA700 gel scanner, and quantified using Fiji.

*Cell volume measurements*. WT cardiac fibroblasts were nucleofected with pCDNA3.1 tdTomato (Neon Transfection kit, Invitrogen, MPIK10025), diluted 1 in 2 with control cells (cell nucleofected with no DNA), and seeded at high density in µslides (IBIDI, 80826). Cells were grown to confluence and entrained with temperature cycles. Twenty-four hours before the start of the recording medium was changed to "Air medium" supplemented with 1% Hyclone III, and plates sealed. Z-stacks were acquired every hour for 2.5–3 days using a Leica SP8 confocal microscope. Imaris software (Oxford Instruments) was used for 3D reconstruction, tracking and determination of cell volume. Cell volume values were then normalized to the mean intensity of tdTomato across time (cell volume/mean intensity). For each dataset, comparison of fit was performed with GraphPad Prism. Data were fitted with either a straight line or a circadian damped cosine wave. When a circadian damped cosine wave equation was preferred, only those cells with a period between 20 and 28 h were considered rhythmic.

## Measurement of the effective diffusion rate of Quantum dots

*Streptavidin–Quantum Dot synthesis*. CdSe/CdS/ZnS core/shell Quantum Dots (QDs) were synthesized using a high-temperature reaction of metal carboxylates and sulfur and selenium precursors in octadecene as described previously[93]. They were transferred into water by ligand exchange with a block copolymer ligand composed of a first block of imidazole acting as anchors onto the QD surface and a second polymer block of sulfobetaine and azido-functionalized monomers[93,94]. This second block provides efficient antifouling properties from the sulfobetaine and is amenable to click chemistry mediated bioconjugation. The resulting QDs were purified using ultracentrifugation on sucrose gradients and ultrafiltration and resuspended in HBS buffer (10 mM HEPES, 150 mM NaCl pH 7.5)[94]. For strep-tavidin-conjugation, 10 nmol streptavidin were functionalized with dibenzylcy-clooctyne (DBCO) using a 3:1 ratio of DBCO-NHS:streptavidin in borate buffer (0.1 M pH 8.0). The proteins were purified from unreacted DBCO-NHS using ultrafiltration (Vivaspin, 10 kDa cutoff, 2 rounds of 10 min, 13000 g), resuspended in HBS and mixed with 1 nmol QDs for 24 h at 4 °C. Bioconjugated QDs were finally purified from unreacted proteins by ultracentrifugation in HBS (2 rounds of 25 min, 120000 g). Before delivery into cells, Streptavidin-QDs were incubated with a 15-fold molar excess of biotinylated Cell Penetrating Poly(disulfide)s (CPDs)[95] overnight at 4 °C in (0.01 M HEPES, 150 mM KCl, pH 7). On the day of the experiments, QD-CPDs conjugates were diluted to a final 1 nM concentration in growth medium and added onto cells (see below). For the experiment in Supplementary Fig. 2b commercial streptavidin-conjugated QDs (Thermo - Q10101MP) were used. The different size and coating of these QDs might account for the different effect size between experiments in Fig.1e and Supplementary Fig. 2b[96].

*Time-course experiment*. WT fibroblasts were seeded in 35 mm dishes (WPI, FD3510-10), grown to confluence and differentially entrained with temperature cycles. QD motion was imaged at predicted trough and peak of protein oscillations, i.e. 24, 36, 48, and 60 h after medium change (DMEM 1% Hyclone III, supplemented with 10 mM HEPES to avoid any change in pH during imaging). Briefly, cells were incubated with 1 nM QD-CPD conjugates for 1 h, washed twice with PBS and imaged at 37 °C, atmospheric conditions. To prevent any perturbations by media change, all treatments were performed at 37 °C using the conditioned medium from the same dishes. For each time-point, we acquired several time-lapse recordings from 2 to 3 independent replicate dishes (between 22 and 43 movies per timepoint per replicate), and replicates were subsequently pooled for analysis.

*In vitro measurements*. For in vitro measurements of quantum dot diffusion, streptavidin-coated quantum dots (Thermo, Q10101MP) were added at a final concentration of 200 pM to solutions of 35kD polyethylene glycol or BSA dissolved at different concentrations in 20 mM Tris pH 7.6 (or as labeled) + 150 mM KCl. Quantum dots were then imaged in imaging dishes (WPI, FD3510) coated for 1 h with 0.1 mg/ml PLL-g-PEG (SuSoS, PLL(20)-g[3.5]- PEG(2), dissolved in 10 mM Hepes pH 7.6). Imaging was performed at 25 °C using the same imaging conditions as imaging of quantum dot diffusion in cells.

*Imaging and image processing*. Imaging was performed by spinning disk confocal microscopy using a custom-built setup based on a Nikon Ti stand equipped with a perfect focus system, a PLAN NA 1.45 100X objective and a spinning disk head (Yokogawa CSUX1). QDs were excited by a 488 nm laser (Coherent OBIS mounted

in a Cairn laser launch) and imaged using a Chroma 595/50 emitter filter. Images were recorded with a Photometrics Prime 95B back-illuminated sCMOS camera operating in high gain mode (12 bits dynamic range), gain 1 and pseudo global shutter mode (synchronized with the spinning disk wheel). Each acquisition consisted of 1000 frames acquired at 62.5 Hz (10 ms exposure, 6 ms readout). The system was operated by Metamorph.

Estimation of the effective diffusion coefficient of QDs was performed in Fiji[89] and MATLAB 2017b (Mathworks) using custom codes available on request. In brief, for each field of view, QD position was precisely determined by 2D Gaussian fitting using the plugin Thunderstorm[97]. Average localization precision across the entire dataset based on photon count was $14 \pm 0.0004$ nm (mean $\pm$ sem $n = 214$, 368, 144 QDs). QD trajectories were then tracked using the MATLAB adaptation by Daniel Blair and Eric Dufresne of the IDL particle tracking code originally developed by David Grier, John Crocker, and Eric Weeks in MATLAB 2017b (http://site.physics.georgetown.edu/matlab/index.html).

For each track, the Mean Square Displacement (MSD) of segments of increasing duration (delay time ($t$) was then computed ($MSD(t) = (\Delta x)^2 + (\Delta y)^2$) using the MATLAB class MSD Analyzer[98]. MSD curves were then fitted to a subdiffusion model captured by the function $MSD(t) = 4Dt^{\alpha}$[99] with $D$ the effective diffusion rate. We then filtered the MSD curves to retain only those with a good fit ($R^2 > 0.9$) and with $\alpha > 0.5$ as we found this was the most accurate way to exclude immobile particles. Indeed, as shown previously[95], QDs delivered by our CPD technology exhibit both immobile and diffusive motion in cells, and immobile QDs have to be filtered out as they would artificially lower the estimated diffusion coefficient. We then accurately estimated the effective diffusion of these filtered tracks by fitting the first 50 points of their MSD curve by the function $MSD(t) = 4Dt$. The median of the effective diffusion rate D was then averaged across all the filtered tracks to give an estimate of the effective diffusion rate for a given field of view (average number of filtered tracks per field of view was always >1000). The effective diffusion rates per field of view were then average for each timepoint of the circadian time-course.

*Inductively coupled plasma-mass spectrometry (ICP-MS) analysis—mammalian cells*. For time-course experiments, cells were seeded at high density and grown to confluence. Cells were entrained for three days with temperature cycles (12 h at 32 °C, 12 h at 37 °C). At the transition to the cold phase of the last cycle, medium was replaced with fresh medium and cells were moved into constant conditions (37 °C). Samples were extracted every 3 h for 3 consecutive days, beginning 24 h after the medium change. Cells were washed once with ice-cold "iso-osmotic buffer 1" (300 mM sucrose, 10 mM Tris base, 1 mM EDTA, pH 7.4 adjusted with phosphoric acid (Sigma, 79614, 330–340 mOsm) and once with ice-cold "iso-osmotic buffer 2" (300 mM sucrose, 10 mM Tris base, 1 mM EDTA, pH 7.4 adjusted with acetic acid Sigma, 45727, 330–340 mOsm). Cells were then lysed with high purity 65% nitric acid (HNO$_3$, Merk Millipore, 100441) supplemented with 0.1 mg/L (100 ppb) cerium (Romil, E3CE#) as a procedural internal standard. Samples were diluted in high purity water to a final concentration of 5% HNO$_3$, then run using a Perkin Elmer Nexion 350D ICP-MS. Data were collected and analysed using Syngistix version 1.1. Ion concentrations were normalized for a scaling factor based on cerium abundance and analysed using GraphPad Prism. Outliers were identified and removed using the ROUT ($Q = 0.1\%$, high stringency) or the Grubbs' (alpha = 0.2) method in GraphPad Prism. In order to determine circadian rhythmicity, data were fit by comparing a straight line with a damped cosine wave equation with the simpler model chosen unless the latter was preferred with $p < 0.05$.

*ICP-MS analysis – heart tissue*. Male C57BL/6 J mice kept under 12 h/12 h-light/dark (L/D) cycles were sacrificed every 4 h, at the indicated zeitgeber times (ZT). Mice were euthanized by cervical dislocation, confirmed by exsanguination. Heart tissues were removed and stored on ice in "iso-osmotic buffer 1" (see above). Tissues were diced into 3 mm slices, washed extensively with ice-cold iso-osomotic buffer 1 then 2, and gently digested with an NP40-based buffer on a tube rotator (50 mM tris pH 7.5, 1% NP40, 5 mM EDTA, 50 mM choline bitartrate). Samples were spun down (21.000 g x 10 min) and the supernatant quantified using BCA (for protein quantification) and analysed by ICP-MS (for ionic composition). For each sample, ion abundance was normalized to protein abundance.

*Kinase assay*. Full-length human WNK1 (residues 1-2382;) was expressed with a C-terminal MYC-DDK (1xFLAG) tag from a pCMV6-Entry backbone (Origene, USA, RC218208) in Expi 293 F suspension cells (Thermo Fisher Scientific, A14527)[100]. Cells were grown at 37 °C, 8% CO$_2$ and shaking in 1 L roller bottles with vented caps. A culture of 300 mL of cells at $2\times10^6$ cells/mL was transfected with a mix consisting of 10 mL Expi 293 expression medium, 900 µL 1 mg/mL PEI MAX 40k (Polysciences, 24765-1) and 300 µg DNA, preincubated at room temperature for 15 min. Cells were harvested 36 hours later and spun down at 1000 g for 5 min. Cell pellets were washed once in PBS and flash frozen. Pellets were thawed and resuspended in 20 mL lysis buffer A (50 mM Hepes pH 7.5, 125 mM KOAc, 5 mM MgAc$_2$, 0.5% Triton X-100, 1 mM DTT, 1X PhosSTOP (Sigma-Aldrich, 4906845001), 1X complete EDTA-free protease inhibitor cocktail (Roche, 11873580001)), and the resulting lysate was centrifuged at 30 min in a SS-34 rotor. The supernatant was mixed with 100 µL anti-FLAG M2 affinity resin (Millipore-Sigma, USA) and incubated at 4 °C for 90 min. The beads were then washed with 9 mL of lysis buffer, 12 mL of wash buffer 1 (50 mM Hepes pH 7.5,

400 mM KOAc, 5 mM MgAc$_2$, 0.1% Triton X-100, 1 mM DTT, 1X complete EDTA-free protease inhibitor cocktail), and finally 6 mL of wash buffer 2 (50 mM Hepes pH 7.5, 100 mM KOAc, 5 mM MgAc$_2$, 1 mM DTT, 10% glycerol). WNK1 was then eluted from the resin in 200 μL wash buffer 2 containing 0.25 mg/mL 3X FLAG peptide.

Recombinant WNK1 kinase domain (residues 1-487) was expressed with a N-terminal His$_{14}$-bdSUMO tag in *E. coli* Rosetta 2 cells (Novagen, 71397). Expression was induced with addition of 0.2 mM IPTG and cells were grown for 8 hours before harvesting. Cell pellets were resuspended via sonication in lysis buffer B (50 mM Tris pH 7.5, 300 mM NaCl, 20 mM imidazole, 5 mM DTT, 1 mM PMSF, 10% glycerol), and the lysate was then centrifuged for 45 min at 1034 g in a SS-34 rotor. The resulting supernatant was mixed with Ni$^{2+}$ affinity agarose resin (Qiagen, 30210) and incubated at 4°C for 60 min. The resin was washed with two column volumes (CV) of wash buffer 3 (50 mM Tris pH 7.5, 300 mM NaCl, 20 mM imidazole) and then with one CV of wash buffer 4 (50 mM Hepes pH 7.5, 300 mM KOAc, 2 mM MgAc$_2$, 10% glycerol, 5 mM DTT). The recombinant WNK1 kinase domain was eluted via addition of 50 nM bdSENDP1 protease (Frey *et al.*, 2014; Addgene ID 104962) in wash buffer 4.

*Recombinant PDPK1* was ordered in a purified form (abcam, ab60834). Kinase assays were performed with 50 ng protein in assay buffer (50 mM Hepes pH 7.5, 100 mM KOAc, 2 mM MgAc$_2$, 0.1 mM ATP, 0.2 mCi/mL γ-$^{32}$P-ATP, 1X Phos-STOP) with the indicated concentrations of either NaCl or PEG 20,000 (Sigma, 81300-1KG). Reactions were incubated at 32°C for 30 min and then quenched with sample buffer and boiled. Samples were then diluted sevenfold prior to electrophoresis due to the high concentrations of PEG. Incorporation of γ-$^{32}$P was analyzed by exposing the gel to a Phosphor screen for 24–72 h. Uncropped blots are presented in Supplementary Fig. 14.

*Microelectrode array (MEA) recordings.* Immediately after isolation, $1 \times 10^5$ cardiomyocytes were seeded in a small drop on the center of the MEA chip (Multi Channel Systems, 60MEA200/30iR-Ti-gr) coated with 10 μg/mL fibronectin (Sigma, F1141). After 5 h, fresh medium was added. After 8 days in culture, with medium change performed every day, a last medium change was performed and cells kept in constant conditions. Field potential from each MEA was then recorded at predicted peaks and troughs of ion rhythms before returning cells to the incubator between time-points. Cells were maintained at 37 °C during transfer to and from the incubator. Recording was done in culturing medium supplemented with 10 mM HEPES using the MEA recording device (MEA2100-2×60-System, Multi Channel Systems). All recordings were performed at atmospheric conditions with stage and custom-built heated lid held at 37 °C. MEAs were allowed to equilibrate for 200 s and local field potentials from all electrodes were recorded at 10 kHz for 10 min. Data were recorded and analysed using the Multi-Channel experimenter and Multi-Channel DataManager software. Data from individual MEAs (several replicates per experiment) are shown in the main figures and in supplementary figures.

*Langendorff heart electrophysiology.* Animals were culled and hearts excised between ZT0-1 or ZT11-12. Excess tissue was removed under dissection microscope in ice-cold Krebs-Henseleit Buffer (KHB) solution (118 mM NaCl, 11 mM Glucose, 1.8 mM CaCl$_2$, 4.7 mM KCl, 25 mM NaHCO$_3$, 1.2 mM MgSO$_4$, 1.2 mM KH$_2$PO$_4$). Hearts were cannulated through the aorta and held in place using fine thread tied around the aortic vessel. The cannula was then attached to a jacketed glass coil filled with oxygenated KHB and heart perfused at 37 °C at 4 ml/min. This whole process took ~3–4 min. Monophasic action potentials were recorded using custom-made silver chloride electrodes from the left atrial appendage. Any hearts that did not stabilize within the first 10 min of recording were excluded from further study. Signals were amplified and digitized using a Powerlab system (4/35) and analysed in LabChart (v8; ADinstruments, UK). After brief electrical stimulation of the right atrial appendage (600 Hz for 15 s), average heart rate was calculated over 30 seconds of stable recording. Mice in the treated group were given rapamycin (dissolved in drinking water) for 2 weeks befre culling.

*ECG telemetry implantation and analysis.* Mice were implanted with telemetry devices (ETA-F20; Data Sciences International, USA) for electrocardiographic recording. Mice were anaesthetized with isoflurane, following which the telemetry remote was inserted into the abdominal cavity. Biopotential leads were secured ~1 cm right of midline at upper chest level (negative) and ~1 cm to the left of midline at the xiphoid plexus (positive). Mice were allowed a recovery period of 10 days before individual housing and the start of ECG recording. 10 second ECG waveform sweeps were collected every 5 min except for during autonomic blockade were ECG was recorded continuously. Following 5 days of ECG recording mice were given rapamycin in drinking water for 2 weeks before 5 more days of recording. At ZT0 or ZT12, mice were given 10 mg/kg metoprolol i.p. followed by 4 mg/kg atropine to achieve total autonomic blockade, confirmed by a sharp and prolonged decrease in heart rate variability (RR interval standard deviation). Intrinsic heart rate was measured over 30 min following atropine injection.

ECG analysis was carried out as previously described[85] using bespoke ECG software written in MATLAB (R2018a; Mathworks, USA). Briefly, for every 10 s sweep $R$ waves were detected by amplitude thresholding and ECG waveforms

compared to the median of the sweep. Any that deviated significantly from this template were removed from subsequent analysis. Any sweeps with low signal/noise ratio or where >20% of detected events were excluded during template matching were excluded. Heart rate was calculated from the median RR interval of each sweep.

*Mice for Langendorff and telemetry studies.* Male, C57BL/6 J mice aged 10–13 weeks at the start of study, purchased from Charles River (UK), were used for ECG and Langendorff studies. Langendorff studies included some Bmal1$^{f/f}$ animals bred in-house on the same C57BL/6 J background. Mice were housed under 12:12 LD (~400/0 lux) at $22 \pm 2$ °C with food and water available *ad libitum*.

*Behavioral experiment for NKCC1 KO mice.* Sperm from NKCC1$^{-/-}$ mice (FVB.Cg-Slc12a2tm1Ges/Mmjax, the Jackson Laboratories) was used for in vitro fertilization of PER2::LUC transgenic mice. After two rounds of breeding within the background strain, NKCC1$^{+/+}$-PER2::LUC and NKCC1$^{-/-}$-PER2::LUC mice ($n = 4$ males per genotype, 8–12 weeks of age) were individually housed, and locomotor activity monitored via wheel-running. Mice were subjected to 12 h/12 h-light/dark cycles (L/D) for 6 days then released into constant darkness (D/D) for 14 days. Period of locomotor activity was determined using ClockLab software (Actimetrics) (last 6 days in D/D).

*Osmometry.* Stock solutions of 600 mg/ml hemoglobin (MP Biomedicals, 02151234), 400 mg/mL PEG-20K (Sigma, 81300), or 400 mg/mL KCl were prepared in PBS. Osmolality was measured using OSMOMAT 30 (Gonotec). Three measurements for each dilution were performed. Data were analysed using GraphPad and fitted with either a straight line or a third order polynomial. All cell media were adjusted to be isoosmotic within each experiment, unless otherwise stated.

*Ratiometric pH measurements.* Native Fluc spectra were collected in solution using a Cary Eclipse Fluorescence Spectrophotometer (Agilent) with 10 nM Quantilum (Promega, Madison, WI; E1701) in 30 mM MOPS, 10 mM β-mercaptoethanol, 1 mM ATP pH 7.4, 1 mM potassium luciferin, 10 mM MgSO$_4$, and 1 mg/mL bovine serum albumin, as described previously[101]. Cellular pH calibration was performed with stably transfected SV40:Luc U2OS cells at 37 °C in white 96 well plates (Corning™ 3917) on a Tecan Spark 10 M, alternately using 550–575 nm and 610–635 nm filters to calculate 560/620 nm ratio (Supplementary Fig. 2c, d) with 5 s integration times. At the depicted timepoint, 10 μL of 10 μM FCCP was automatically injected to give a final concentration of 1 μM FCCP. Longitudinal measurements were also performed on a Tecan Spark 10 M (Supplementary Fig. 2c, d), under the same conditions, measuring total bioluminescence from each well (1 s integration) in addition to emission at 550–575 and 610–635 nm from the same well (5 s integration). For these experiments cell media was buffered with 20 mM MOPS instead of bicarbonate, as described previously[88]. The pH of the various media was measured and adjusted at 37 °C with 1 M HCl or NaOH, and then adjusted to 342 mOsm with 1 M NaCl.

*Drugs.* DIOA (Sigma, D129) was used at 50 μM, bumetanide (Sigma, B3023) at 100 μM, torin1 (Merck, 47599) at 50 nM, FCCP at 0.3 μM, and closantel (Sigma, 34093) at 30 μM. Rapamycin (J62473, Alfa Aesar, UK) was dissolved at 50 mg/ml in ethanol, then diluted 1:1000 in water. Rapamycin solutions were kept shielded from light using tin foil and changed weekly.

*Statistical analysis.* Statistical analyses were performed using GraphPad Prism. Data are presented as mean ± SEM unless ostherwise stated. Statistical significance was determined using two-tailed student's $t$ test or ANOVA, as appropriate. Post hoc tests were used to correct for multiple comparisons as indicated in figure legend. p-values are reported using the following symbolic representation: ns $p > 0.05$, *$p \leq 0.05$, **$p \leq 0.01$, ***$p \leq 0.001$, ****$p \leq 0.0001$.

All animal experiments were licensed under 1986 Home Office Animal Procedures Act (UK) and carried out in accordance with local animal welfare committee guidelines.

**Reporting Summary.** Further information on research design is available in the Nature Research Reporting Summary linked to this article.

## Data availability
The data supporting the findings of this study are available in the main article and supplementary files or from the corresponding author upon reasonable request. Source data are provided with this paper as Source Data File. The original mass spectra and search engine files used in this study have been deposited in the public proteomics repository MassIVE as MSV000087673 [https://massive.ucsd.edu/ProteoSAFe/dataset.jsp?accession=MSV000087673]. Source data are provided with this paper.

## Code availability
The codes used detection, tracking, and analysis of quantum dots are available for download through GitHub (https://github.com/deriverylab/Stangherlin2021).

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

## Acknowledgements
We thank Alex Harmer, Helen Causton, and past and present O'Neill lab members for valuable discussion and contribution, particularly Priya Crosby and Ned Hoyle, visual aids, and the biological services group for assistance with animal work and husbandry. Funding: E.D. was supported by the Medical Research Council (MC_UP_1201/13) and the Human Frontier Science Program (CDA00034/2017-C); RSE by a Wellcome Trust Sir Henry Dale Fellowship (208790/Z/17/Z). This work was supported by the AstraZeneca Blue Skies Initiative and the Medical Research Council (MC_UP_1201/4).

## Author contributions
J.S.O., P.N., R.S.E., and A.S. conceived the idea and wrote the manuscript. E.B and S.M. synthesized the cell penetrating poly(disulfide)s, T.P. and N.P. synthesized quantum dots. A.S., J.W., D.W., S.B., A.Z., E.S., S.P.C., A.B., E.H., A.G, A. I., M.P., J.D., R.V., D.B., E.D., and R.S.E. performed experiments and analysis, G.v.O. made pivotal intellectual contributions.

## Competing interests
P.N. holds shares in AstraZeneca. The other authors declare no competing interests.
