## [Peer Review File · Nature Communications]

REVIEWER COMMENTS

Reviewer #1 (Remarks to the Author):

The manuscript by Stangherlin et al. seeks to understand the what are the consequences for having such a strong circadian proteome. Many reports now demonstrate that up to 20% of the circadian proteome is circadian (this % is likely to increase when higher time resolution studies are performed). Circadian control of mTOR activity is causing some of this circadian proteome. Indeed circadian proteostasis is likely to become a very significant topic across many fields. Given that the osmotic pressure exerted by changes in the soluble proteome must be compensated, the authors investigate whether this occurs via differential control of ion transport, and if so what is the consequence of this. The authors propose that increased molecular crowding and low cellular chlorine activate the WNK/OXSR1 pathway to phosphorylate slc17A family members to balance the osmostasis with postassium, chloride and sodium ions. The authors find that this daily variation of ion flux plays a role in cardiac muscle firing in-vivo.

Overall, this is a very interesting paper which reveales new fundamental processes which will be highly relevant across many systems. A range of quite complex experiments have been conduced to a very high standard using some very novel state of the art techniques that provides insight into circadian changes observed within cells that consistently manage their osmotic homeostasis throughout the day. Overall this paper is of sufficient standard for publication in Nature Communication. However, there are areas of the paper that require further investigation. The author's mechanism involves the slc17A family, however, this is only addressed in a broad superficial manner. There are numerous members of the slc17A family, however, all are treated as one "family" and individual roles are not investigated. The paper relies heavily on small molecule inhibition and more investigation at the genetic level could help to rule out off target effects to strengthen the confidence in the mechanism.

More specific concerns are listed below:

1. Fig 2E – The authors acknowledge the potential for the small molecules used to inhibit slc12A to target other ionic transporters. They have also found from phosho-proteomic databases that slc12A2, slc12A5 and slc12A6 are phosphorylated in a circadian manner. Do these 3 transporters play a more dominant role? This the contribution of each be teased out further with genetic manipulation of these genes, such as, the use of siRNA, which should be feasible at least in the fibroblast model.
2. Fig 2C – The authors use tau mice that have a decreased period of their rhythms. The

presentation of results makes it difficult to differentiate the difference between WT and tau mice. Are these rhythms lost in cells that lack a functional clock, a *Bmal1*^{-/-} cell line for instance may have more stark differences and would provide more evidence for the role of the molecular clock in osmostasis. The authors direct us to another paper from the group that is on BioRxiv which illustrates effects with cryptochrome on proteostasis, but a genetic perturbation of the clock and demonstration of its effects would significantly strengthen the findings of this paper.

3. Fig3 – This figure proposes that molecular crowding leads to the activation of the WNK/OXSR1 pathway to activate the *slc17A* family resulting in net potassium efflux or ionic change? However, potassium levels, or any ionic levels, are not measured in any experiment in the figure following manipulation of WNK/OXSR1 activity levels. Does adding the bolus of serum, used in fig 2, activate the WNK/ OXSR pathways similarly to PEG used in this figure?

4. Fig 3E, inhibition of OXSR1 leads to higher soluble protein. Does this imply that OXSR1 upstream and is regulating mTOR and protein synthesis? Furthermore, these effects should be confirmed by genetic ablation of WNK or OXSR1 to rule out any off-target effects of the inhibitor used. If the authors knockdown WNK/OXSR1 and add a bolus of serum, one would predict that there would be an increase in protein synthesis but no change in ion levels?

Minor comments:

1. There should be discussion around how the system resets itself. A lot is discussed about increased in molecular crowding due to increased mTOR activity and this increases *slc12A* transport of ions but there should be some discussion about it's return to baseline.
2. In the results section referring to the antiphase oscillations, it should be made clearer in the text as to the direction of travel of the ions. Currently, it is just referred to potassium and sodium transport but whether it is transported into or out of the cell should be included.
3. The cell type should used should be included in the figure legends.
4. In figure 2, fig 2a is mislabelled as fig 2b.

Reviewer #2 (Remarks to the Author):

Stangherlin and colleagues report that electroneutral active transport of Na⁺, K⁺, and Cl⁻ by SLC12 transporters compensate osmotic alteration in cultured fibroblasts and In

cardiomyocytes during daily protein rhythms and speculate about possible molecular determinants of these regulatory pathways. The manuscript is based on interesting observations, but appears - in its present form – incomplete and suffers from methodological limitations.

1. The authors describe cell-autonomous rhythm in the abundance of soluble proteins, but not of membrane bound proteins. Why only soluble proteins and why not also membrane proteins? How is this distinct regulation organized? Reading this part of the manuscript it came to my mind that changes in ion concentrations might not be the compensation, but part of a regulatory process compensated by changes in soluble protein expression.

2. For membrane transport not the absolute amount, but the concentration of ions represents the thermodynamically significant value. The authors should provide changes in ion concentrations and describe how they oscillate during the day.

3. I would assume that changes in neuronal ion concentrations might contribute to sleep awake rhythm, whereas changes in heart excitability will be compensated by the autonomic nervous system. I do not understand why the authors spend so much effort on cardiac cells, and not on the nervous system.

4. Drugs blocking SLC 12 transporters have been in clinical use since decades. I never heard about changes in daily rhythms in these patients. If the here reported processes are of major physiological importance, some side effects should have been reported. Moreover, KO animals lacking NKCC1 should have major impact on these processes, and the authors should test this.

5. P. 8. "Our observations suggest that mTORC-dependent increases in cytosolic protein are initially...". Cells exhibit a marked variability in $[Cl^-]$, ranging from values close to 100 mM to very low values. If this hypothetical regulatory pathway occurred, distinct cells should react in a variable manner.

Reviewer #3 (Remarks to the Author):

In this manuscript research is described that studies the circadian variation of soluble

protein concentration in the cell. It is shown that this rhythmicity is controlled by mTORC1 activity that by itself is following a circadian rhythm. In synchronized fibroblasts it is shown that both the peak of soluble protein abundance and mTORC1 activity is shifted by approximately 12 hours to the peak of PER2 activity (as evidenced by a PER2 dependent Luciferase assay). The abundance of soluble protein concentrations correlates with cytosolic translation. A diffusion rate assay further supports that soluble proteins content and molecular crowd correlate. As cell volume is not under circadian rhythm, it is postulated that there must be compensatory mechanisms. It is shown that the ionic cytosolic concentrations of the most prominent ions Na, K and Cl follow an opposite rhythm and depend on the activity of SLC12A co-transporters (NKCC2, and KCC). Moreover, inhibition of these transporters interferes with the rhythmicity of soluble protein concentration. Evidence is provided that the regulation of these compensation is provided by the WNK/OXSR1 pathway. It is then shown that these mechanisms are also important in cardiomyocytes and in vivo in the heart.

This is a nice manuscript, most of the experiments are elegant and well done. However, I would have liked that the authors address a few more questions:

1) Their work suggests that the WNK-OXSR1 pathway is involved in the control of the compensatory mechanisms adjusting ions, and also involving NKCC and KCC. According to their in vitro kinase assays they demonstrate, by using different concentrations of PEG that molecular crowding stimulates WNK1 activity. What is the mechanism? I am not aware of it, but it must be a direct effect. With this assay it should be possible to get some insights how this is working.

2) In Fig.1b and d it is shown that the circadian variation is not seen in total cellular protein extract, and, surprisingly, that this is reciprocal in nuclear/organelle fraction. The authors speculate that this reciprocity may be part of a mechanism to sequester cytosolic proteins to other organelles, when necessary (at time points when soluble proteins are low). The authors should demonstrate that during these periods (e.g. 24h or 48h), there is indeed accumulation of cytosolic proteins in the organelles.

3) The authors are just providing an experiment combining the 2 inhibitors against NKCC and KCC that these proteins are involved in the opposite rhythm of Na, K, Cl as compared to the soluble proteins. The authors should show the contribution of the 2 transporters individually by applying the inhibitors individually.

Other comments:

Figure 2. a is mislabeled on the figure (top left), likely also in Fig. 2e and f, as the y axis is not labeled similarly in these 2 figures.

In general, for all the concentrations (proteins, ions), the authors should provide in the legend absolute values (for example to what concentration corresponds 100%).

All the protein gels should show molecular weight markers. In

Fig. 3C, for example it is not obvious to the reader which band of the p-OSXR1 blot corresponds to total OXSR1. Moreover, the blot with the characterization of p-OSXR1 antibody is not convincing. Despite treatment of the extract with phosphatase the p-OSXR1 are still visible. If the antibody is specific for the phosphorylated form, the bands should completely disappear, as phosphatases are generally very efficient. Hence this has to be improved, to convince us that we are really dealing with a specific antibody.

Further in Fig.3b, in this assay we are looking at autophosphorylation of WNK1. For the average reader in Nature Communication this is not necessarily obvious and should be stated.

The author should also state some where in the paper that WNK1 mutations contribute to an autosomal genetic disease, called pseudohypoaldosteronism type II (also referred to as familial hyperkalemic hypertension (FhHt) or Gordon's syndrome).

Olivier Staub

Reviewer #4 (Remarks to the Author):

The authors of this manuscript show diurnal rhythm of intracellular soluble protein concentration that is regulated by oscillatory activation of mTORC1 in mouse fibroblasts, which supposedly cause rhythmic change of osmotic pressure and subsequent alteration of cell volume. However, no such rhythm in cell size is observed. Further studies have revealed that reciprocal oscillations of ion concentrations (K⁺, Na⁺ and Cl⁻) regulated by SLC12A-OXSR1 axis compensate osmotic change induced by mTORC1-mediated enhancement of soluble protein concentration. Moreover, the authors find this machinery plays an important role in circadian regulation of electrical activity in cardiomyocytes. The

work as a whole is of broad interest for biology research areas about circadian modulation of intracellular of osmotic homeostasis and cell volume. But several points (major concerns) should be addressed before the publication of this manuscript.

Major concerns:

1. The authors suggested that the balance between soluble protein and ion concentrations minimized circadian change of cell volume, as there was no significant rhythm observed in cell size of the majority of mouse fibroblasts they measured (Fig2a and suppl. Fig.3a).

However, a recent study (as mentioned in the legend of suppl. Fig.3a) has shown diurnal oscillations in both sizes and protein levels of hepatocytes in which mTORC1 activity is also rhythmic, which is not logically compatible with the hypothesis of the current study.

2. In this study, the authors used inhibitors of mTORC1, SLC12A, and OXSR1 in experiments to address questions related to diurnal changes (for example, Fig2f, 3e; suppl. Fig2a, 2b). Considered the half life, side effect and toxicity of these drugs and the very long exposure times in those experiments (for example, up to 60 hour treatment in Fig2f), the data of those experiments are not convincing. To provide evidence directly supporting the hypothesis, it would require studies using genetic models in which signaling via mTORC1 pathway, or SLC12A-OXSR1 axis has been abrogated.

3. mTORC1-mediated increase of soluble protein concentration should be due to the enhancement of de novo protein synthesis that would cause the decrease of intracellular amino acid concentrations, which could also compensate the increased osmotic pressure. The authors should design experiments to explore this possibility and compare it with the compensatory effect of SLC12A-OXSR1 axis.

Minor comments:

1. What is the reason that causes the constant decrease of the average cell volume of fibroblasts as shown in Fig2a? The authors should check the vitality of these cells.

2. In the experiments of Fig2d, 2e, and 2f, it should be better to use amino acids to stimulate cells, since 10% serum contains lots of other nutrients and growth hormones.

3. Fig2a was mislabeled as Fig2b.

Reviewer #5 (Remarks to the Author):

General:

As I understand it, the authors claim to experimentally validate their hypothesis that osmotic balance in living cells is maintained throughout the circadian cycle by dynamic adjustment of the ionic composition of the cellular interior to compensate for time-dependent change in the concentration of intracellular protein. They have performed measurements of the time-dependence of intracellular apparent viscosity by tracking the motion of tracer particles and claim that this is a measure of 'macromolecular crowding'. In parallel they measure the intracellular ionic composition as a function of time and report that the intracellular concentrations of Na, K and Cl are in antiphase with time-dependent changes in total protein concentration and 'macromolecular crowding', and assert that this relationship underlies maintenance of osmotic balance and cell volume.

This is an extremely complex MS because it reports a large number of different experiments carried out in multiple laboratories, and interpretations of those experiments. Descriptions of the data and analysis are quite condensed, and without sufficient detail to enable others to validate the results. Most of the experimental results are presented in the form of figures of highly processed rather than raw data.

Specific comments:

1. The authors should be aware that the rates and extents of biochemical reactions are governed by the concentrations of reactants, not dimensionless relative amounts scaled to some arbitrary value such as the maximum value. This observation is crucially important when it is reported that cellular volume is decreasing by 30 -50% over the time course of the experiments (Fig 2a, erroneously labeled 2b) while total protein content is roughly constant (Fig 1b). Thus the total intracellular protein concentration is increasing by 30 - 50% over the time course of the reaction. Since the initial total protein concentration already exceeds several hundred mg/ml, a further increase in total concentration by 30 - 50% may alter the rates and extents of biochemical reactions by as much as several orders of magnitude (Minton, chap 10 of Cellular and Molecular Physiology of Cell Volume Regulation (K Strange, ed., CRC Press, 1994; Zhou et al, Ann Rev Biophys 37: 375; 2008). These changes in absolute concentration must not be ignored.

2. Why is the cellular volume decreasing substantially over the time course of the experiments? Does this not indicate that the volume regulatory mechanisms of the cells being studied are seriously impaired? If so, how can the authors maintain that osmotic balance is maintained over the duration of the experiment, which is at the heart of their hypothesis?

3. The authors state (top of MS p.5) that the level of macromolecular crowding in a solution is inversely proportional to the rate of diffusion of a macromolecule in that medium. I don't know what the authors mean by 'macromolecular crowding' since they never define it, and I can find no references to articles on macromolecular crowding. While the rate of macromolecular diffusion of a particular species in a crowded medium does depend upon the fraction of total volume occupied by macromolecules (one element of crowding) in the medium, it depends on many other factors as well, none of which appear to be considered in this MS. The authors' statement is highly simplistic and qualitatively inaccurate.

4. The postulated inversely proportional relationship between the rate of diffusion of quantum dots in cells and total protein concentration must be examined in more detail. This is certainly not predicted theoretically nor has it been observed experimentally in model systems (Muramatsu & Minton, PNAS 85:2984; 1988). In any event, the results reported here are inconsistent with this hypothesis. We can calculate the relative total concentration of soluble protein at 24, 36, 48, and 60 hr by dividing the relative amount at these time points indicated in Fig 1b by the relative volume indicated in Fig 2a, erroneously labeled 2b (average of values from 3 curves). In the Figure reproduced below we plot the measured apparent diffusion coefficient at each of the time points against the inverse of the total protein concentration calculated as described above.

These results do not support the postulated relationship between diffusion coefficient of quantum dots and soluble protein concentration.

The only proper way to establish a quantitative relationship between the diffusion of quantum dots and the composition of the intracellular medium in a medium as complex as cytoplasm is to prepare lysates of known total concentration, ion composition, and pH, to measure the rate of diffusion of quantum dots and the viscosity, and the dependence of both quantum dot diffusion rate and viscosity in the lysate upon protein concentration, ion composition, and pH. This is clearly a

separate research project of its own and cannot be lumped together with reports of numerous other experiments.

5. Expression of different proteins depends upon time in the circadian cycle.

Because intracellular protein concentration is so high, changes in protein composition are likely to result in time-dependent changes in intracellular pH, an additional modulator of intermolecular interactions in the soluble phase. pH changes would contribute to time-dependent changes in the state of association and in the solubility of individual proteins, all of which affect the osmotic pressure. The effect of pH change does not appear to have been considered in this work.

6. The authors' data show that the total solubility of protein is time-dependent. There is a large literature on the dependence of protein solubility upon environmental factors such as the concentration and composition of other macromolecules, osmolytes, and ions, as well as the pH. None of this literature appears to have been consulted. An understanding of these time-dependent changes depends upon understanding the interactions between the proteins and their environment.

7. The authors have not considered the very large deviations from thermodynamic ideality that exist in complex media comparable to cytosol. In the 1990's Parker and colleagues showed that mechanisms of volume regulation via compensatory ion transport were extremely sensitive to very small changes in intracellular volume and the concentration of intracellular protein. They proposed that this sensitivity was due to large changes in thermodynamic activity associated with small fractional change in intracellular protein concentration when that concentration is hundreds of mg/ml (Colclasure & Parker, *J Gen Physiol* 100: 1; 1992; Parker & Colclasure, *Mol Cell Biochem* 114:9-11; 1992; Minton et al, *PNAS* 89:10504; 1992). The authors should examine and discuss the relationship between this work and the present work.

We thank the reviewers for their comments and for providing insightful suggestions. We trust that the point by point rebuttal letter below will address all their queries.

Reviewer: *italic*

Authors: plain text

REVIEWER COMMENTS

Reviewer #1 (Remarks to the Author):

The manuscript by Stangherlin et al. seeks to understand the what are the consequences for having such a strong circadian proteome. Many reports now demonstrate that up to 20% of the circadian proteome is circadian (this % is likely to increase when higher time resolution studies are performed). Circadian control of mTOR activity is causing some of this circadian proteome. Indeed circadian proteostasis is likely to become a very significant topic across many fields. Given that the osmotic pressure exerted by changes in the soluble proteome must be compensated, the authors investigate whether this occurs via differential control of ion transport, and if so what is the consequence of this. The authors propose that increased molecular crowding and low cellular chlorine activate the WNK/OXSRI pathway to phosphorylate slc17A family members to balance the osmostasis with postassium, chloride and sodium ions. The authors find that this daily variation of ion flux plays a role in cardiac muscle firing in-vivo.

Overall, this is a very interesting paper which reveales new fundamental processes which will be highly relevant across many systems. A range of quite complex experiments have been conduced to a very high standard using some very novel state of the art techniques that provides insight into circadian changes observed within cells that consistently manage their osmotic homeostasis throughout the day. Overall this paper is of sufficient standard for publication in Nature Communication. However, there are areas of the paper that require further investigation. The author's mechanism involves the slc17A family, however, this is only addressed in a broad superficial manner. There are numerous members of the slc17A family, however, all are treated as one "family" and individual roles are not investigated. The paper relies heavily on small molecule inhibition and more investigation at the genetic level could help to rule out off target effects to strengthen the confidence in the mechanism.

We thank the reviewer for engaging with the scope of our work and for the positive comments.

We regret if we inadvertently implied that SLC12A family members function equivalently to each other, it was in fact SLC12A *family members* that we focused on (and not SLC17A). In the revised manuscript, we now stress the differential functions of SLC12A members at multiple points. Excepting neuron-specific KCC2, we do treat KCC transporters as being functionally equivalent and similarly regulated, because this is broadly established in the literature²⁻⁵. We also now include new experiments that address the differential functions of NKCC and KCC transporters both in unsynchronised cells and during the circadian cycle (Fig. 3e, Supplementary Fig. 7a). The results of these experiments are consistent with predictions from our model. We also now include genetic evidence from NKCC1 knockout/knockdown cells (Supplementary Fig. 7b, Supplementary Fig. 10-11). We trust that the additional experiments in our revised manuscript provide sufficient evidential support to assuage the reviewer's concern.

Regarding small molecule inhibitors, in our original manuscript, we focused on small molecules rather than genetic approaches for three reasons. First, genetic deletion affects the steady state osmolarity (Supplementary Fig. 10), not its dynamics. Daily dynamics are the focus of the mechanism described in this paper, one that is already very well established in the context of regulatory volume increase/decrease (RVI/D). Second, osmotic homeostasis is robustly defended due to redundancy between SLC12A family members and other RVI/D mechanisms because it is essential for cell viability – any genetic manipulation that completely abolished RVI/D would result in cell death. For reasons outlined in the revised discussion, we do not propose SLC12A transporters are the only relevant mediators of compensatory ion transport, they are simply a ubiquitous and physiologically important one. Third, transient pharmacological inhibition allows us to overcome the stated limitations of genetic approaches to thereby explicitly test the specific hypothesis that inhibition of NKCC or KCC will elicit differential effects on circadian ion transport that depend upon the biological time of treatment (New Figure 3e).

We do agree that these points should be emphasized more strongly and have added additional genetic data and discussion to the revision of our manuscript. For example, the revised manuscript now provides an explicit justification for why we do not think that a deeper attempt to explore mechanism by classical genetic approaches would be likely to provide further insight into the mechanistic underpinnings of compensatory ion transport, as follows:

Line 280: “The model we propose for N(K)CC-mediated compensatory ion transport pivots around dynamic daily changes in WNK-OSXR1/SPAK1 activity and differential KCC vs NKCC activity, and primarily informed by acute inhibition of signalling and transporter function within a circadian cycle, rather than across it. The molecular mechanisms of SLC12A regulation are already well-characterised in the context of RVI/D, following extracellular osmotic challenge^{17,20,61,66–68}, we simply propose that RVI/D regulation of NKCC1 vs KCC extends to encompass intracellular osmotic challenge over the circadian cycle without significant volume change.

The SLC12A family includes 9 members (including KCC1-4) whose overlapping and semi-redundant functions are important for physiological function and overall viability^{37,69,70}. The same is true for the 4 WNK paralogs, as well as OSXR1 with SPAK1^{31,68}. This combination of overall essentiality coupled with functional redundancy between components, presents a barrier to the use of genetic approaches for delineating the specific contribution of individual transporters and kinases over the circadian cycle. NKCC1 deletion is tolerated however, apparently through upregulation of other (non-SLC12A) transporter activities⁷¹.”

More specific concerns are listed below

1. Fig 2E - The authors acknowledge the potential for the small molecules used to inhibit slc12A to target other ionic transporters. They have also found from phospho-proteomic databases that slc12A2, slc12A5 and slc12A6 are phosphorylated in a circadian manner. Do these 3 transporters play a more dominant role? This the contribution of each be teased out further with genetic manipulation of these genes, such as, the use of siRNA, which should be feasible at least in the fibroblast model.

We focused on the SLC12A family of transporters due to their established role in regulatory volume increase/decrease, and because their observed phosphorylation rhythm was concordant with the ion rhythms we observed. A technical limitation of these circadian phosphoproteomics studies is that a circadian rhythm, or lack thereof, can only be determined for phospho-peptides that are sufficiently abundant to be detected at all time points. Thus, there is no reason to believe that activity rhythms of other SLC12A members do not occur, simply that other SLC12A phosphopeptides were not detected. Moreover, SLC12A family members are unlikely to be the

only transporters involved in the compensatory mechanism we describe, but rather we just show that they contribute to it. We clarify this important point in the discussion of the revised manuscript, as follows:

Line 383: “SLC12A members are not the only transporters that facilitate RVI/D under acute challenge^{17,76}, and so are unlikely to be the only effectors of compensatory circadian solute transport. Nor is K⁺ the only cellular osmolyte of relevance to osmotic compensation, it is merely the most abundant (by an order of magnitude). Moreover, yeast cells, which lack close homologues of SLC12A family transporters, but have OSXR1/SPAK orthologs, show the same antiphasic rhythms of K⁺ and soluble protein that we observed in mouse fibroblasts⁴⁵. Thus, whilst we do not discount that other transporters, such as LRRC8 family members⁷⁶, contribute to circadian ion fluxes, our observations indicate that SLC12A-mediated electroneutral ion transport is an important mitigator of the osmotic challenge presented by daily rhythms in soluble protein abundance; thereby allowing cytosolic composition to change each day whilst defending osmotic homeostasis and cell volume. Because the transport of small osmolytes buffers intracellular osmotic potential against changes in cytosolic macromolecule concentration, the cellular capacity for K⁺-export likely confers daily variation upon the capacity of cells to respond to exogenous stimuli through changes in protein expression.”

In fibroblasts, we can exclude any contribution from NKCC2 and KCC2 since they are not expressed. SLC12A8 and SLC12A9 are uncharacterised^{5,6}, whereas we were unable to identify sufficiently specific antibodies with which knockdown of KCC1, KCC3 and KCC4 could be distinguished and validated. However, the extensive literature on the overlapping and semi-redundant function of KCC transporters in the context of RVI/D suggests there is no reason to believe that knockdown of one KCC would not be compensated by the increased activity of the other two⁶. However, we were able to validate knockdown of NKCC1/SLC12A2, and we hope the reviewer is satisfied that our observations that were entirely consistent with the effect of pharmacological NKCC1 inhibition (Supplementary Fig. 7a, b).

Moreover, we also performed a series of experiments capitalizing on the NKCC1 knock out mouse model (Supplementary Fig.10). We used this model since NKCC1 is the sole isoform of the ubiquitously expressed sodium/potassium/chloride transporter (the NKCC2 isoform is selectively expressed in the kidney). We first analysed the ionic content of NKCC1^{-/-} mouse lung fibroblasts. We found that NKCC1^{-/-} cells have less Na⁺ and K⁺ than their NKCC1^{+/+} counterparts. Importantly, NKCC1^{-/-} cells have more protein (both soluble and total), indicating that they have adapted to the lack of NKCC1 and they have an altered homeostatic set point for the cytosolic protein/ion equilibrium. While the proportional response to 10 % serum in terms of cytosolic protein was unaffected, NKCC1^{-/-} cells showed a blunted activation of S6 kinase and reduced phosphorylation of OXSR1. Furthermore, we found that ablation of NKCC1 has a profound effect on cellular and organismal circadian rhythms (Supplementary Fig. 11) as delineated in the response to referee #4 point 2. This is now communicated in the text as follows:

Line 289: “The SLC12A family includes 9 members (including KCC1-4) whose overlapping and semi-redundant functions are important for physiological function and overall viability^{37,69,70}. The same is true for the 4 WNK paralogs, as well as OSXR1 with SPAK1^{31,68}. This combination of overall essentiality coupled with functional redundancy between components, presents a barrier to the use of genetic approaches for delineating the specific contribution of individual transporters and kinases over the circadian cycle. NKCC1 deletion is tolerated however, apparently through upregulation of other (non-SLC12A) transporter activities⁷¹. Consistent with previous reports^{69,71}, NKCC1^{-/-} mice showed obvious growth and developmental defects (small size, deaf, unsteady gait) and weaned in non-Mendelian ratios. Fibroblasts derived from adult PER2::LUC NKCC1^{-/-} mice had significantly reduced cellular Na⁺ and K⁺ levels with commensurately increased soluble and

total protein levels (Supplementary Fig. 10a). Whilst NKCC1 deletion clearly perturbs steady state osmostasis^{31,37}, rather than its daily dynamics, this genetic evidence further validates the reciprocal relationship between cellular ion and macromolecular content, and indicates that NKCC1^{-/-} cells adapt to the loss of this transporter through a permanently altered set point for osmotic homeostasis. Importantly, NKCC1^{-/-} cells showed a similarly proportional increase in soluble protein following serum stimulation with respect to wild type, which our model predicts to be initially KCC-mediated, and demonstrates that NKCC1^{-/-} cells remain competent to facilitate an increase in soluble protein levels from an altered baseline (Supplementary Fig. 10b). Critically however, NKCC1^{-/-} cells showed blunted mTORC1 and OXSR1 activation following serum-stimulation (Supplementary Fig. 10c), consistent with feedback from osmostasis to regulate the means of macromolecular synthesis, and consistent with previous observations^{45,71}. Interestingly, NKCC1^{-/-} knockout cells and mice also exhibited lengthened period of circadian rhythms in cellular gene expression and mouse locomotor activity (Supplementary Fig. 11). This is not especially surprising, since mTORC1 signalling regulates circadian rhythms⁸, and mTORC1 activation is regulated by the capacity for effecting compensatory ion efflux, which will be less efficient without NKCC1 to facilitate Cl⁻ recycling according to our model (Supplementary Fig. 9c).”

We do understand the reviewer’s request for additional genetic evidence to tease apart the relative contribution of each SLC12A family member. As discussed above, we do not think that this is not the appropriate test of our hypothesis that the daily dynamics of overall KCC vs NKCC1 activity facilitate circadian changes in the cytosolic soluble protein. Certainly, the effect of NKCC1 deletion is supportive of our model, but please note that the primary effect of gene deletion is to perturb steady state osmotic homeostasis (Supplementary Fig. 10), which affects the circadian rhythm itself (Supplementary Fig. 11), making any comparison with the wild type difficult to interpret. Whereas, acute inhibition of KCC vs NKCC1 activity within a circadian cycle allows us to test the time-of-day dependent variation predicted by our model, and could not readily be achieved by genetic approaches (revised Fig. 3e).

2. Fig 2C - The authors use tau mice that have a decreased period of their rhythms. The presentation of results makes it difficult to differentiate the difference between WT and tau mice. Are these rhythms lost in cells that lack a functional clock, a Bmal1^{-/-} cell line for instance may have more stark differences and would provide more evidence for the role of the molecular clock in osmostasis. The authors direct us to another paper from the group that is on BioRxiv which illustrates effects with cryptochrome on proteostasis, but a genetic perturbation of the clock and demonstration of its effects would significantly strengthen the findings of this paper.

We appreciate the referee's concern, which gives us the opportunity to clarify a crucial aspect of our findings. In the BioRxiv paper the reviewer mentions¹, we used *cry1/cry2* homozygous knock out cells as model of dysfunctional transcriptional clock. Under the current paradigm, cells that lack CRYPTOCHROME function should not display any circadian rhythms. However, we found that whilst CRY-deficient cells lack rhythms in *Per2* mRNA, they exhibit circadian rhythms in PER protein production⁷, as well as both soluble protein and ion abundance¹. This finding adds to an increasing body of evidence that suggests that gene expression rhythms normally facilitate but are not required for some elements of circadian cell function. Indeed, cell-autonomous circadian rhythms have been reported in *Cry*-deficient and *Bmal1*-deficient cells^{1,7-10}, which lack the "core clock genes" that are necessary for gene expression rhythms. Our observations¹ rather suggest that "clock gene"-mediated cycles of transcriptional feedback suppress daily cycles of protein abundance to maintain protein homeostasis, rather than generate them. In support of this, the amplitude of cytosolic protein rhythms is increased in *Cry*-deficient cells, that lack transcriptional rhythms, with a compensatory increase in the relative amplitude of cellular K^+ rhythms (Referee Figure 1).

Figure 1. Circadian rhythms in soluble protein and K^+ persist in the *Cry1/Cry2* knock out model (CKO). (a) From one time-course experiment, ions, cytosolic proteins and total protein were extracted in parallel samples. Blue lines highlight the antiphasic relationship between oscillations in cytosolic soluble protein and potassium abundance. Mean \pm SEM, p-values from RAIN, red lines are fits by a damped cosine compared with a straight line (null hypothesis). Parallel *PER2::LUC* recordings were also performed and plotted as a phase marker. (b) Relative amplitudes of cytosolic protein and potassium concentrations oscillations in (a) were greater in CKO compared to WT (Student's t test with Welch correction, mean \pm SD).

Thus, genetic perturbation of clock gene activity does result in a clear phenotype, although not the one predicted by the current TTFL paradigm. Therefore, using *Bmal-* or *Cry*-null models, where translational rhythms persist, is not the ideal model with which to test the hypothesis that rhythms in ion transport are under cell-autonomous circadian regulation. To accomplish this, we employed the *tau* mutation, which shortens the period of oscillation. Using this model, a similar shortening of period was observed for all three measurements (Supplementary Fig. 6c, *PER2::LUC*, cytosolic protein, ion abundance), whilst their respective phase relationships were unchanged.

The combined data from *tau* mutant and *Cry*-deficient models provide strong genetic evidence that ion and protein rhythms are generated by cellular timekeeping mechanisms, and do not require CRYPTOCHROME. This point is not pursued in the discussion since it is not germane to our findings, but is addressed in Wong *et al.*, which we hope will be published soon.

3. Fig3 - This figure proposes that molecular crowding leads to the activation of the *WNK/OXSRI* pathway to activate the *slc17A* family resulting in net potassium efflux or ionic change? However, potassium levels, or any ionic levels, are not measured in any experiment in the figure following manipulation of *WNK/OXSRI* activity levels. Does adding the bolus of serum, used in fig 2, activate the *WNK/OXSRI* pathways similarly to PEG used in this figure?

The referee makes an excellent suggestion for which we are most grateful. As requested we applied a bolus of serum and measured OXSR1 phosphorylation. Similar to low Cl⁻ media, a bolus of serum stimulates to OXSR1 phosphorylation (Fig. 3c). Please note that in the revised manuscript we also confirm that a bolus of serum induces a time-dependent change in the cytosolic crowding measured using quantum dots (Supplementary Fig. 3c).

In addition, to address the referee's concern about the lack of K⁺ measurements, we have measured cytosolic K⁺ levels upon acute inhibition of KCC and NKCC (Fig. 3e), at the peak and trough of the ion/protein oscillation. These data show an opposite effect of NKCC vs KCC inhibition on K⁺ levels, consistent with their differential activity in the regulation of K⁺ ion fluxes. The major finding of this manuscript is that ion rhythms buffer osmostasis against circadian variation in protein abundance; we believe that this set of data further validates our model and thank the reviewer again for suggesting these experiments.

4. Fig 3E, inhibition of OXSR1 leads to higher soluble protein. Does this imply that OXSR1 upstream and is regulating mTOR and protein synthesis? Furthermore, these effects should be confirmed by genetic ablation of WNK or OXSR1 to rule out any off-target effects of the inhibitor used. If the authors knockdown WNK/OXSR1 and add a bolus of serum, one would predict that there would be an increase in protein synthesis but no change in ion levels?

As the reviewer suggests, over the course of hours OXSR1 is indirectly both upstream and downstream of mTORC, as it is an essential mechanism for protein and osmotic homeostasis i.e. OXSR1 activity is required for NKCC1 activity, which ultimately limits the capacity for sustained mTORC1 activation. Similar to the issue with KCC1, KCC3, and KCC4 (discussed above), the reviewer highlights a general problem in studying proteins, such as histones or tubulin, whose function is essential for cell viability and is therefore encoded by multiple paralogous genes, which are therefore semi-redundant. There are 4 WNK genes with overlapping function, whereas OSXR1 functions redundantly with SPAK1. A wealth of prior observations suggests that any combination of knockdowns which is sufficiently efficacious to affect overall WNK1-4 or OXSR1/SPAK function will result in cellular stress, which will be epistatic to the biological question¹¹. This is why we have primarily used time as a variable, rather than genetic manipulations that affect steady state osmostasis i.e. the activity of these proteins over acute timescales is already well-characterised, our data suggest that their activities vary over daily timescales, and thus we test the prediction that the same manipulation will produce a different outcome dependent on the biological time that the perturbation occurs – see Figure 3e for example. To satisfy the reviewer's concern we have performed experiments under acute serum stimulation (Fig. 3c, Supplementary Fig. 10c). Our data suggest that OXSR1 phosphorylation is mTORC-mediated, indicating that mTORC is upstream in the pathway and consistent with previous studies¹². In addition, we also now show that less protein accumulates upon serum stimulation when KCC or NKCC are inhibited (Supplementary Fig. 7a), consistent with our model that the capacity for ion transports determines the scope for changes in cytosolic soluble protein abundance.

The reviewer's final prediction is not really consistent with our model or observations. In the RVI/D literature and our model, KCC is basally active by default and functions to facilitate the osmotic equilibrium over the plasma membrane, whereas WNK-OSXR1/SPAK signalling facilitates KCC inactivation not activation. Consistent with this, we find that inhibition of KCC abolishes mTORC-dependent increase in soluble protein (Supplementary Fig. 7a). In addition, whether or not an increase in soluble protein is observed depends on the biological time the

stimulus is encountered (Fig. 2e). Thus, if serum stimulation occurs when mTORC1 activity and soluble protein is already high, then no additional increase in protein is expected (Fig. 2e). If serum stimulation occurs when mTORC1 activity is usually low, however, then the SLC12A-mediated ion export that directly compensates for the soluble protein increase triggered by mTORC activation should be KCC-mediated (Fig. 3e, left panel) and so will not be directly affected by OSXR1/SPAK or WNK inhibition.

Minor comments:

1. There should be discussion around how the system resets itself. A lot is discussed about increased in molecular crowding due to increased mTOR activity and this increases slc12A transport of ions but there should be some discussion about it's return to baseline.

We appreciate the referee's comment. In order to make this point clearer, we have edited the text as follows:

Line 264: "This phase is followed by the circadian decrease in mTORC activity, resulting in gradual reduction of cytosolic protein. The osmotic equilibrium over the plasma membrane is now maintained by increased in import of Na⁺, K⁺ and Cl⁻ via NKCC and other transporters, with KCC now being much less active."

2. In the results section referring to the antiphase oscillations, it should be made clearer in the text as to the direction of travel of the ions. Currently, it is just referred to potassium and sodium transport but whether it is transported into or out of the cell should be included.

We thank the reviewer for the suggestion we have modified the text as follows:

Line 179: "Importantly, we observed ~24 h ion rhythms for intracellular K⁺, Na⁺ and Cl⁻, which oscillated in antiphase with soluble protein (Fig. 2b, Supplementary table 2): ions are exported when soluble protein increases and imported when soluble protein decreases. Several other biologically-relevant ions were not similarly rhythmic (Supplementary Fig. 6b)."

3. The cell type should used should be included in the figure legends.

We now include the cell type used in the figure legends.

4. In figure 2, fig 2a is mislabelled as fig 2b.

We apologize for the mistake, which has now been corrected.

Reviewer #2 (Remarks to the Author):

Stangherlin and colleagues report that electroneutral active transport of Na⁺, K⁺, and Cl⁻ by SLC12 transporters compensate osmotic alteration in cultured fibroblasts and In cardiomyocytes during daily protein rhythms and speculate about possible molecular determinants of these regulatory pathways. The manuscript is based on interesting observations, but appears - in its present form - incomplete and suffers from methodological limitations.

We thank the reviewer for the positive feedback. We trust that the new experiments address these concerns.

1. The authors describe cell-autonomous rhythm in the abundance of soluble proteins, but not of membrane bound proteins. Why only soluble proteins and why not also membrane proteins? How is this distinct regulation organized? Reading this part of the manuscript it came to my mind that changes in ion concentrations might not be the compensation, but part of a regulatory process compensated by changes in soluble protein expression.

This investigation addresses osmotic homeostasis and so we were primarily interested in soluble cytosolic proteins because, compared membrane proteins, they make a much larger contribution to cytosolic osmotic potential. We observed no significant variation in total protein during these experiments, whereas the amount of protein that can be extracted from cells by digitonin lysis (Fig. 1b) or NP-40/fractionation (Fig. 1d) exhibits daily variation. As the reviewer will observe in Fig. 1d, there is clear reciprocal variation in the amount of protein in the nuclear/organelle fraction, and so it is entirely plausible that this represents the rhythmic sequestration then liberation from organelles as we recently observed in yeast cells¹³. Whether a similar mechanism to the one we identified over the yeast respiratory oscillation also operates over the circadian oscillation in mammalian cells is the subject of a separate investigation, which will be communicated in due course. In contrast, this manuscript explores the cellular consequences of these soluble protein rhythms rather than their generation. We address this in the text, as follows:

Line 133: “Whether daily cycles of cytosolic protein levels are primarily attributable to rhythmic protein synthesis/degradation or rhythmic protein sequestration/liberation warrants further investigation. However, we can exclude pH-dependent changes in cytosolic protein solubility, since no variation in cytosolic pH was detected over the cellular circadian cycle (Supplementary Fig. 2).”

The second point raised by the referee is interesting, and we pondered it too. Indeed, as a part of a separate follow up study, we are investigating the dynamics of circadian protein expression at the membrane. However, we believe that this work is also beyond the scope of the current manuscript.

Figure 2. Protein changes are not driven by ion changes. Soluble protein upon a bolus of serum in normal medium or Na-free medium (1 h incubation, cardiac fibroblasts).

Regarding the reviewer’s last point, in a cycle it is always difficult to separate cause from effect and, as we show, the interaction between ion transport and soluble protein is bidirectional. However, when cell media is replaced with iso-osmotic, Na-free media, we actually find the soluble protein increase is attenuated not accentuated (see reviewer Figure 2). This argues for the ion changes being driven by protein changes but not the other way around, whereas the capacity for ion export clearly limits the extent to which soluble protein can increase during the circadian cycle (Fig. 2 d-e, Fig. 7a, c).

2. For membrane transport not the absolute amount, but the concentration of ions represents the thermodynamically significant value. The authors should provide changes in ion concentrations and describe how they oscillate during the day.

We thank the referee for raising this important question. The average concentration for Na⁺, K⁺, and Cl⁻ is now presented in Supplementary Table 2. We also present absolute values for ions and protein rhythms in Supplementary Fig. 1d, Supplementary Fig. 6b, and Supplementary Fig. 12a.

3. I would assume that changes in neuronal ion concentrations might contribute to sleep awake rhythm, whereas changes in heart excitability will be compensated by the autonomic nervous system. I do not understand why the authors spend so much effort on cardiac cells, and not on the nervous system.

Figure 3. Circadian rhythms in intracellular Na⁺ and K⁺ in hypothalamic neurons. Time course showing Na⁺ and K⁺ abundance in primary hypothalamic neurons over time (n=3). Ion abundance was determined by Inductively coupled plasma - mass spectrometry. Data are presented as mean±SE. p-values indicate comparison of damped cosine wave with straight-line fit (null hypothesis = no rhythm).

This is a fantastic suggestion and one that we are actively pursuing. Indeed, we have preliminary data showing ion rhythms in cultured primary hypothalamic neurons (Reviewer Figure 3). However, both cell types are irreplaceable, terminally-differentiated somatic cells, and cardiomyocytes are no less important to human health than are neurons. In our experience, however, primary cardiomyocytes offer the experimental virtues of reduced heterogeneity and increased robustness compared with primary neurons. We therefore preferred this model to test key predictions from our model. We are now pursuing our neuron findings using human iPS neurons, rather than neonatal mouse, since they offer vastly improved inter-experimental reproducibility and will communicate them in due course.

The referee is correct in that *in vivo* the heart rate is regulated by the autonomic nervous system. However, the functional contribution made by timing mechanisms that are intrinsic to heart are now becoming understood¹⁴. Indeed, our data show that ion oscillations and time of day variation in firing rate occur *in vitro*, in the absence of autonomic regulation. As mentioned in our discussion, this mechanism likely represents a cellular circadian anticipation of the increased workload that is placed upon the heart during an organism's active phase, and may have crucial implications for recipients of heart transplants, that lack vagal innervation. Indeed, behavioural and pharmacological approaches could be used to entrain heart cells and improve the daily variation in heart rate, in order to optimise cardiac output of these patients. In the accompanying accepted manuscript (Hayter *et al.*) a more thorough investigation of the relative contribution of heart-intrinsic vs autonomic control of cardiac electrophysiology has been carried out, and some striking findings about the underlying basis of the daily rhythm in susceptibility to arrhythmia are reported.

4. Drugs blocking SLC 12 transporters have been in clinical use since decades. I never heard about changes in daily rhythms in these patients. If the here reported processes are of major physiological importance, some side effects should have been reported. Moreover, KO animals lacking NKCC1 should have major impact on these processes, and the authors should test this.

To our knowledge, the long-term effects of diuretics on circadian rhythms have not been investigated *in vivo*. Whether or not an effect would be expected would depend on the serum half-life of the drug. In most circumstances the administration of diuretics aims actually at

restoring circadian rhythms in renal physiology. In the case of high blood pressure for example, it is now well accepted that the best pharmacological effect of diuretics is achieved when drugs are administered before bedtime, to reduce blood pressure during the night¹⁵.

As suggested by the referee, we analysed the effect of genetic ablation of NKCC1 on circadian rhythms in primary cells and in mice (Supplementary Fig. 11). We found that circadian rhythms of clock gene expression are impaired in NKCC1^{-/-} fibroblasts, and display a significant but modestly lengthened period of oscillation. In line with this, we also observed a longer circadian period of locomotor activity in NKCC1^{-/-} mice. For mammals, the suprachiasmatic nucleus (SCN) of the hypothalamus is the master synchroniser of circadian rhythms throughout the body, for which daily light-dark cycles is the dominant timing cue. The reviewer may be interested to learn that SLC12A transporters are thought to be critical to the SCN-encoding of photoperiod that underlies seasonal responses in mice, and that *ex vivo*, SCN timekeeping is indeed sensitive to NKCC1 and KCC antagonists¹⁶, but at concentrations that would not be achieved *in vivo* due to the blood-brain barrier.

5. P. 8. “Our observations suggest that mTORC-dependent increases in cytosolic protein are initially...”. Cells exhibit a marked variability in [Cl⁻], ranging from values close to 100 mM to very low values. If this hypothetical regulatory pathway occurred, distinct cells should react in a variable manner.

The referee raises a good point. Most of this regulatory pathway is already well-established in the context of RVI/D, and we simply propose its function extends to encompass the maintenance of cell volume and osmolarity against internal change in osmotic potential in addition to external changes. The reviewer is also absolutely correct that stochastic cell-to-cell variability in Cl⁻ levels has been observed *in vivo*, using fluorescent chloride reporters¹⁷ by our colleague Gian Michele Ratto. It is something that we hope to extend to circadian cellular assays in future, but accurate, non-destructive longitudinal chloride measurement is a long-standing problem which is currently hampered by the challenge of cell tracking (which requires exposures every 30-60 minutes), low signal-to-noise ratio (which requires long integrations or high laser power) and photo-toxicity (which requires short and infrequent exposures with low laser power for cells to remain healthy over 3 days in culture). Sadly, the ICP-MS method used for chloride detection in the present manuscript is not sufficiently sensitive to allow single-cell measurements. Thus, whilst we agree with the reviewer’s prediction, to test it lies beyond what is technically feasible with current methods. We hope the reviewer will agree that this has no direct bearing on our model, which is informed by population level measurements, and provides important context and hypotheses for whenever the technology allows the reviewer’s prediction to be tested experimentally.

Reviewer #3 (Remarks to the Author):

In this manuscript research is described that studies the circadian variation of soluble protein concentration in the cell. It is shown that this rhythmicity is controlled by mTORC1 activity that by itself is following a circadian rhythm. In synchronized fibroblasts it is shown that both the peak of soluble protein abundance and mTORC1 activity is shifted by approximately 12 hours to the peak of PER2 activity (as evidenced by a PER2 dependent Luciferase assay). The abundance of soluble protein concentrations correlates with cytosolic translation. A diffusion

rate assay further supports that soluble proteins content and molecular crowd correlate. As cell volume is not under circadian rhythm, it is postulated that there must be compensatory mechanisms. It is shown that the ionic cytosolic concentrations of the most prominent ions Na, K and Cl follow an opposite rhythm and depend on the activity of SLC12A co-transporters (NKCC2, and KCC). Moreover, inhibition of these transporters interferes with the rhythmicity of soluble protein concentration. Evidence is provided that the regulation of these compensation is provided by the WNK/OXSRI pathway. It is then shown that these mechanisms are also important in cardiomyocytes and in vivo in the heart.

This is a nice manuscript, most of the experiments are elegant and well done.

We thank the reviewer for the positive feedback.

However, I would have liked that the authors address a few more questions:

1) Their work suggests that the WNK-OXSRI pathway is involved in the control of the compensatory mechanisms adjusting ions, and also involving NKCC and KCC. According to their in vitro kinase assays they demonstrate, by using different concentrations of PEG that molecular crowding stimulates WNK1 activity. What is the mechanism? I am not aware of it, but it must be a direct effect. With this assay it should be possible to get some insights how this is working.

The referee is correct, the effect of PEG on WNK activity is direct and has been elucidated in a recent crystallographic study¹⁸. In this paper the authors show that osmolytes (and PEG) affect WNK structure and activation, which we have confirmed with our *in vitro* assay (Fig. 3b). We have added this important reference in the text.

2) In Fig. 1b and d it is shown that the circadian variation is not seen in total cellular protein extract, and, surprisingly, that this is reciprocal in nuclear/organellar fraction. The authors speculate that this reciprocity may be part of a mechanism to sequester cytosolic proteins to other organelles, when necessary (at time points when soluble proteins are low). The authors should demonstrate that during these periods (e.g. 24h or 48h), there is indeed accumulation of cytosolic proteins in the organelles.

As the referee pointed out this is indeed a speculation. Using differential centrifugation, we detected a time of day partitioning of proteins between the cytosolic fraction and the organellar/nuclear fraction. This does not necessarily mean that cytosolic proteins translocate to organelles. Another possibility is that they partition with non-membrane-bound compartments, also known as biomolecular condensates, which would be in line with our recent findings in yeast¹³ where we found that proteins and enzymes are sequestered into ribonucleoprotein granules to facilitate temporal organisation of cell physiology. Understanding the mechanism that underlies daily rhythms in cytosolic soluble protein is an active line of investigation in our lab, but is far from complete, and has no direct bearing on the current study which explores the consequences of these protein rhythms rather than their generation. To communicate this more clearly, we have modified the sentence on page 5 as follows:

Line 129: “While total protein levels did not change, we observed reciprocal time-of-day variation in protein abundance between the cytosolic and nuclear/organellar fractions. This is compatible with a cell-autonomous daily rhythm in the sequestration of cytosolic proteins to other cellular compartments or membrane-less organelles, which has been observed in several model

organisms^{13,19,20}. Whether daily cycles of cytosolic protein levels are primarily attributable to rhythmic protein synthesis/degradation or rhythmic protein sequestration/liberation warrants further investigation.”

3) The authors are just providing an experiment combining the 2 inhibitors against NKCC and KCC that these proteins are involved in the opposite rhythm of Na, K, Cl as compared to the soluble proteins. The authors should show the contribution of the 2 transporters individually by applying the inhibitors individually.

We apologise if this point was not clear. The purpose of this experiment was to test the hypothesis that ion transport is required to accommodate acute changes in soluble protein. The experiment was done in unsynchronised cells, meaning that the cells were not at specific phase of the circadian cycle. Therefore, to maximise the effect of ion transport inhibition we applied both inhibitors at the same time. To address the referee's comment, we have applied the two inhibitors individually and we found that selective inhibition of either KCC or NKCC attenuated the accumulation of cytosolic protein (Supplementary Fig. 7a), and is consistent with our model that KCC is required for net K⁺ export when soluble protein is increased, but that some NKCC1 activity is also required to provide Cl⁻ to serve as a counterion.

The results of the experiment suggested by reviewer 1 were also consistent with our model, that differential KCC and NKCC activity facilitate net fluxes of K⁺ in opposite directions at different times of day i.e. time-of-day variation in the effect of NKCC and KCC inhibition on cytosolic K⁺ abundance (Fig. 3e).

Other comments:

Figure 2. a is mislabeled on the figure (top left), likely also in Fig. 2e and f, as the y axis is not labeled similarly in these 2 figures. ???

We apologise for the confusion. The panels have been relabelled: the Y axis in panel 2e (now Fig. 2d) has been relabelled as “Soluble protein (% of control)” and panel 2f (now Fig. 2e) as “Soluble protein (% change)” because this is the % change as compared to DMSO at each time point (acute stimulation).

In general, for all the concentrations (proteins, ions), the authors should provide in the legend absolute values (for example to what concentration corresponds 100%).

We now provide the absolute values for ion and protein concentrations in Supplementary Fig. 1d, 6b, 12a and imputed average values in Table 2.

All the protein gels should show molecular weight markers. In Fig. 3C, for example it is not obvious to the reader which band of the p-OXR1 blot corresponds to total OXR1.

We apologize for the lack of clarity. As indicated in the figure legend the correct band for phospho-OXR1 was indicated by a star and the full blot with molecular markers was presented in supplementary figure 8 (now Supplementary Fig. 14). To make this clearer we have now added the molecular weight marker to the representative blots and indicated the correct band with a star also in the full blot.

Please note that during the revisions of the paper, the anti-OXSR1 antibody ab125468 (Abcam) was discontinued. All the relevant samples have been rerun and probed with another anti-OXSR1 antibody (ab224248, Abcam). The representative immunoblots have been replaced.

Moreover, the blot with the characterization of p-OXSR1 antibody is not convincing. Despite treatment of the extract with phosphatase the p-OXSR1 are still visible. If the antibody is specific for the phosphorylated form, the bands should completely disappear, as phosphatases are generally very efficient. Hence this has to be improved, to convince us that we are really dealing with a specific antibody.

We appreciate the referee's criticism. Dephosphorylation was done on lysates after extraction with RIPA buffer (supplemented with phosphatase and protease inhibitors), and was not efficient. We now present phosphorylation of OXSR1 not only upon treatment with hypotonic low chloride solution but also upon serum stimulation, which indicates that we are monitoring the correct band.

Further in Fig.3b, in this assay we are looking at autophosphorylation of WNK1. For the average reader in Nature Communication this is not necessarily obvious and should be stated.

This is a good suggestion. We labelled the figure and modified the text on page 8 as follows:

Line 237: "We tested and validated predictions from the current consensus model with a kinase assay for WNK autophosphorylation *in vitro* and with cellular assays for OSXR1 activity (Fig. 3b, c, respectively)."

The author should also state some where in the paper that WNK1 mutations contribute to an autosomal genetic disease, called pseudohypoaldosteronism type II (also referred to as familial hyperkalemic hypertension (FhHt) or Gordon's syndrome.

We have now added the following sentence:

Line 92: "Importantly, inactivating mutations of genes encoding SLC12A family members, WNK paralogs, and OSR1/SPAK underlie a wide range of inherited human diseases including pseudohypoaldosteronism type II, Gitelman and Bartter syndromes, highlighting the importance of the WNK-OXSR1/SPAK-SLC12A pathway for normal physiology^{3,21-23}."

Reviewer #4 (Remarks to the Author):

The authors of this manuscript show diurnal rhythm of intracellular soluble protein concentration that is regulated by oscillatory activation of mTORC1 in mouse fibroblasts, which supposedly cause rhythmic change of osmotic pressure and subsequent alteration of cell volume. However, no such rhythm in cell size is observed. Further studies have revealed that reciprocal oscillations of ion concentrations (K⁺, Na⁺ and Cl⁻) regulated by SLC12A-OXSR1 axis compensate osmotic change induced by mTORC1-mediated enhancement of soluble protein concentration. Moreover, the authors find this machinery plays an important role in circadian regulation of electrical activity in cardiomyocytes. The work as a whole is of broad interest for biology research areas about circadian modulation of intracellular of osmotic

homeostasis and cell volume. But several points (major concerns) should be addressed before the publication of this manuscript.

We thank the reviewer for the comments and for recognising the relevance of our work.

Major concerns:

1. The authors suggested that the balance between soluble protein and ion concentrations minimized circadian change of cell volume, as there was no significant rhythm observed in cell size of the majority of mouse fibroblasts they measured (Fig2a and suppl. Fig.3a). However, a recent study (as mentioned in the legend of suppl. Fig.3a) has shown diurnal oscillations in both sizes and protein levels of hepatocytes in which mTORC1 activity is also rhythmic, which is not logically compatible with the hypothesis of the current study.

We appreciate the reviewer's comment about the Sinturel *et al.* study²⁴. Please note that Sinturel *et al.* analysed cytosolic protein rhythms in mouse liver under specific feeding conditions, therefore, as also stated in their manuscript, the rhythms they observed were driven by environmental cycles rather than cell-autonomous circadian rhythms: "*The data presented here suggest that hepatocyte size and global RNA and protein levels oscillate in a daily manner in the mouse liver and that these rhythms are driven by both feeding- fasting and light-dark cycles.*". Sinturel *et al.* also state that no similar change in cytosolic protein or mass was observed in cells from any other tissue, reflective of a tissue specific phenomenon, that occurs only in liver, *in vivo*, under specific feeding conditions. Finally, their investigation rests on the assumption that a change in cell area (measured in 2D) correlates directly with a change in volume (3D), but this was not tested by any additional orthogonal method. In conclusion, we do not think that our data are in contrast with those published by Sinturel *et al.* since we focus on a cell autonomous mechanism, whereas they focus on non-cell autonomous aspects that they propose are specific to feeding-induced changes in hepatic physiology. Indeed, if all mammalian cells showed a similar synchronized rhythm in cell volume, this would surely be evident macroscopically.

2. In this study, the authors used inhibitors of mTORC1, SLC12A, and OXSRI in experiments to address questions related to diurnal changes (for example, Fig2f, 3e; suppl. Fig2a, 2b). Considered the half life, side effect and toxicity of these drugs and the very long exposure times in those experiments (for example, up to 60 hour treatment in Fig2f), the data of those experiments are not convincing. To provide evidence directly supporting the hypothesis, it would require studies using genetic models in which signaling via mTORC1 pathway, or SLC12A-OXSRI axis has been abrogated.

Here the reviewer appears to confuse diurnal with circadian - all the relevant experiments in this manuscript except Fig. 4b, e (heart tissue), and Fig 4f (*in vivo*) were performed in isolated cells under constant conditions. To be clear, SLC12A, WNK, and OSXR1 inhibitors were not employed *in vivo*, and therefore their hepatic clearance/detoxification (half-life) is not relevant.

We would like to clarify that in Fig. 2e, f (now 2d, e), SLC12A and mTORC inhibition was performed acutely, 4 h before each timepoint, not over days. We thank the reviewer for pointing out that this was not clear to him/her, and now report the duration of each treatment in the respective figure legend. In order to show that the concentrations of drugs were not toxic, even with chronic treatment at the concentrations employed, we now include bioluminescence recordings where gene expression rhythms were measured upon treatment with NKCC (bumetanide), KCC (DIOA), and OXSRI (closantel) inhibitors for several days (referee Figure

4). We also provide bioluminescence recordings for torin1 treatment. Where a damping of the oscillation was observed, we provide wash off data after each drug treatment to demonstrate that they were not cytotoxic.

Figure 4. Chronic inhibition of *SLCC12A* transporters and *OXSRI* does not affect cell viability. Bioluminescence recordings showing gene expression rhythms (*PER2::LUC* reporter) upon inhibition of *OXSRI* (closantel), *KCC* (*DIOA*), *NKCC* (bumetanide), or *mTORC* (torin1).

The reviewer is correct that cells and mice were treated continuously with torin 1 and rapamycin for several days and weeks, respectively. Given their widespread research and human clinical applications, we consider that the sustained efficacy and reversibility of both drugs is so well established as to render a specific discussion of this point unnecessary. To satisfy the reviewer's concern, the sustained activity of torin 1 in mouse cells is validated in Supplementary Figure 4b. The stability of rapamycin in aqueous solution was established by²⁵, whereas its pharmacodynamics and pharmacokinetics when orally delivered in mice are reported here²⁶.

We appreciate the reviewer's suggestion to perform genetic experiments. We now address this point directly in the revised manuscript:

Line 289: "The *SLC12A* family includes 9 members (including *KCC1-4*) whose overlapping and semi-redundant functions are important for physiological function and overall viability^{3,27,28}. The same is true for the 4 *WNK* paralogs, as well as *OSXR1* with *SPAK1*^{23,29}. This combination of overall essentiality coupled with functional redundancy between components, presents a barrier to the use of genetic approaches for delineating the specific contribution of individual transporters and kinases over the circadian cycle."

To satisfy his/her request we also now provide a series of experiments using *NKCC1*^{-/-} models (Supplementary Fig. 10-11, please also see response to Referee#1 point 1 and Referee#2 point 4). We found that *NKCC1*^{-/-} fibroblasts have a different ion to protein ratio as compared to WT cells, and that they show a reduced *OXSRI* response to serum. In addition, we show that the lack of *NKCC1* affects circadian rhythms in gene expression and mouse circadian behaviour, as follows:

Line 294: "NKCC1 deletion is tolerated however, apparently through upregulation of other (non-*SLC12A*) transporter activities³⁰. Consistent with previous reports^{27,30}, *NKCC1*^{-/-} mice showed obvious growth and developmental defects (small size, deaf, unsteady gait) and weaned in non-Mendelian ratios. Fibroblasts derived from adult *PER2::LUC* *NKCC1*^{-/-} mice had significantly reduced cellular Na^+ and K^+ levels with commensurately increased soluble and total protein levels (Supplementary Fig. 10a). Whilst *NKCC1* deletion clearly perturbs steady state osmolarity^{3,23}, rather than its daily dynamics, this genetic evidence further validates the reciprocal relationship between cellular ion and macromolecular content, and indicates that *NKCC1*^{-/-} cells adapt to the loss of this transporter through a permanently altered set point for osmotic homeostasis. Importantly, *NKCC1*^{-/-} cells showed a similarly proportional increase in soluble protein following serum stimulation with respect to wild type, which our model predicts to be initially *KCC*-mediated, and demonstrates that *NKCC1*^{-/-} cells remain competent to facilitate an increase in soluble protein levels from an altered baseline (Supplementary Fig. 10b). Critically however, *NKCC1*^{-/-} cells showed significantly reduced *mTORC1* and *OXSRI* activation following serum-stimulation

(Supplementary Fig. 10c), consistent with feedback from osmolarity to regulate the means of macromolecular synthesis, and consistent with previous observations^{13,30}. Interestingly, NKCC1^{-/-} knockout cells and mice also exhibited lengthened period of circadian rhythms in cellular gene expression and mouse locomotor activity (Supplementary Fig. 11). This is not especially surprising, since mTORC1 signalling regulates circadian rhythms³¹, and mTORC1 activation is regulated by the capacity for effecting compensatory ion efflux, which will be less efficient without NKCC1 to facilitate Cl⁻ recycling according to our model (Supplementary Fig. 9c).”

3. *mTORC1-mediated increase of soluble protein concentration should be due to the enhancement of de novo protein synthesis that would cause the decrease of intracellular amino acid concentrations, which could also compensate the increased osmotic pressure. The authors should design experiments to explore this possibility and compare it with the compensatory effect of SLC12A-OXSRI axis.*

We appreciate the referee’s comment. Intracellular amino acids, as well as other organic osmolytes, do contribute to the osmotic potential, and are of relevance for the cellular response to acute osmotic insults^{32–34}. We do not discount that changes in intracellular free amino acids occur following rhythms in protein synthesis. However at the whole cell level, their concentration is trivial compared with K⁺ (<1 mM versus 150 mM)^{35–37}. Moreover, most free amino acids are thought to be stored in the lysosome^{38,39} and so make a particularly small contribution to the osmotic potential of the cytosol compared with K⁺ or other bona fide organic osmolytes such as betaine and choline^{32,34,40–42}.

Figure 5. Comparison of amino acid abundance between whole cell and cytosolic extracts. LCMS-coupled metabolomics analysis of intracellular amino acids in whole cell lysates (solubilised with RIPA buffer) and cytosolic extracts (solubilised by digitonin).

In support of this, we have measured the abundance of several amino acids (LCMS-coupled metabolomics) in whole cell lysates and in the cytosolic fraction (digitonin extraction). The abundance of most amino acids in the cytosolic fraction is around 20% lower than that found in whole cell extract (Referee Figure 5).

Based on published data^{35–37} and our metabolomics results, we anticipate that the contribution of amino acids to the intracellular osmotic potential is minimal. Moreover, please note that we do not claim that the increase in soluble protein concentration is solely attributable to *de novo* protein synthesis. As shown by our data (Fig. 1d) and explicitly stated in the accompanying text, rhythmic partitioning of proteins between cytosol and other organelles is very likely to contribute to the soluble protein rhythms we observe, as we have recently shown in yeast¹³.

Minor comments:

1. *What is the reason that causes the constant decrease of the average cell volume of fibroblasts as shown in Fig2a? The authors should check the vitality of these cells.*

The referee raised a good point. The cell volume experiment was performed on transiently transfected fibroblasts. The apparent decrease in cell volume was due to a decrease in

fluorescent intensity over time through photobleaching. This adversely affected the intensity-based threshold for the 3D reconstruction and thus the quantification of cell volume. We apologise for this oversight and now present the data corrected for photobleaching (Fig. 2a). We would like to emphasise that, despite rhythmic changes in intracellular macromolecule content, there is no significant circadian variation in cell volume across time. Please see also response to Reviewer#5, specific comment 2.

2. *In the experiments of Fig2d, 2e, and 2f, it should be better to use amino acids to stimulate cells, since 10% serum contains lots of other nutrients and growth hormones.*

We thank the referee for the suggestion. The purpose of the experiment was to increase cytosolic protein synthesis by a (pseudo-)physiological means, not to selectively activate mTORC. We employed a serum pulse following serum starvation since this is a well-established means of activating protein synthesis, and the associated increase in soluble protein is abolished by mTORC inhibition. To further validate that this treatment elicits an acute increase in cytosolic soluble protein we have added new data (Supplementary Fig. 3c) showing a time-dependent change in cytosolic crowding upon serum stimulation. Our data suggest that mTORC-mediated increases in protein synthesis contribute to cytosolic protein rhythms, but we do not exclude that other mechanisms, such as partitioning in membraneless-organelles, contribute as well. Also please note that whilst it is true that amino acid starvation inhibits mTORC activity via Rag GTPases⁴³, additional amino acids do not further stimulate mTORC since they are always in excess in standard mammalian cell culture media including ours.

3. *Fig2a was mislabeled as Fig2b.*

We apologise for the mislabelling; we have now corrected. Thanks!

Reviewer #5 (Remarks to the Author):

Comments on Stangherlin et al

General:

As I understand it, the authors claim to experimentally validate their hypothesis that osmotic balance in living cells is maintained throughout the circadian cycle by dynamic adjustment of the ionic composition of the cellular interior to compensate for time-dependent change in the concentration of intracellular protein. They have performed measurements of the time-dependence of intracellular apparent viscosity by tracking the motion of tracer particles and claim that this is a measure of ‘macromolecular crowding’. In parallel they measure the intracellular ionic composition as a function of time and report that the intracellular concentrations of Na, K and Cl are in antiphase with time-dependent changes in total protein concentration and ‘macromolecular crowding’, and assert that this relationship underlies maintenance of osmotic balance and cell volume.

We thank the reviewer for this feedback. We apologise for our poor communication. From what they have written we have the impression that we have failed to adequately convey a couple of key points, as well as the aspects of our study that are novel compared with those that simply validate established principles of cell physiology, as follows:

- We did not actually measure viscosity - the resistance of liquids to deformation. We measured the effective diffusion coefficient of cytosolic nanoparticles - previously established as a reasonable proxy for a known effect that macromolecular crowding has in eukaryotic cytosol, which is to retard the diffusion of similarly sized molecules^{44,45}). As the reviewer is aware, the concept of ‘macromolecular crowding’ is not a new one^{46,47}, being simply a term used to communicate the way in which the concentration of macromolecules affects the activity of macromolecules, as well as the bulk solvent. Our observation that protein concentration and effective diffusion coefficient show an inverse (but non-linear) relationship in cells and in vitro should not be surprising, since it is entirely concordant with theory and prior literature^{46,48,49} – Supplementary Fig. 3).
- Intracellular ion concentration oscillates in antiphase with cytosolic soluble (not total) protein concentration and macromolecular crowding, without volume change. Over the last several decades, the experimental evidence that informs current understanding of the relationship between cell volume and osmotic equilibrium over the plasma membrane (RVI/D) is so strong and well-accepted that we take it as foundational knowledge upon which to build further insight. As cited by the reviewer below, for example, the experimental evidence used to propose the model by Minton et al, PNAS, 1992⁴⁷ suggests the observed rhythm in cytosolic protein concentration (or macromolecular crowding) should elicit rhythmic volume change. There is no circadian volume change, i.e. no net movement of water, but intracellular osmotic potential must be in equilibrium with extracellular osmotic potential, since mammalian cells lack cell walls present in plant and fungal cells. In this case, some other osmotically active species must leave the cell for osmotic homeostasis to be maintained. K⁺ is the major cellular osmolyte (~140 mM), and its rhythm in abundance is opposite to that of cytosolic soluble protein. We do not assert that this relationship underlies maintenance of osmotic balance and cell volume, but simply that we have excluded all other plausible possibilities that are compatible with current understanding of mammalian cellular physiology. We then test this hypothesis and find further evidence to support it during acute assays, meaning that it should now be accepted until experimentally refuted or an equally plausible and testable alternative is proposed.

In our revised manuscript, we have made numerous textual changes to more clearly distinguish established principles and mechanisms of cellular physiology from our own observations and interpretations, for example:

Line 280: “The model we propose for compensatory ion transport pivots around dynamic daily changes in WNK-OSXR1/SPAK1-SLC12A activity, primarily informed by acute inhibition of signalling and transporter function within a circadian cycle, rather than across it. The molecular mechanisms of SLC12A regulation are already well-characterised in the context of RVI/D, following extracellular osmotic challenge^{2,29,50-53}, we simply propose that RVI/D regulation of NKCC1 vs KCC extends to encompass intracellular osmotic challenge over the circadian cycle without volume change.”

This is an extremely complex MS because it reports a large number of different experiments carried out in multiple laboratories, and interpretations of those experiments. Descriptions of the data and analysis are quite condensed, and without sufficient detail to enable others to validate the results. Most of the experimental results are presented in the form of figures of highly processed rather than raw data.

There are no novel methods in this manuscript. As per journal guidelines, rather than provide a detailed protocol for each experiment we state the information that any individual, with prior experience of a given technique, would require to repeat our experiments. In cases where new techniques have been developed within our labs, we have already published detailed protocols which are referenced in this manuscript^{45,54,55}. We are of course happy to describe our methodology in greater detail on request but would like to reassure the reviewer they need have no concern that a competent experimentalist could not validate any or all of our observations from the methods information provided.

All raw data used to generate each figure will be available for download from the journal upon publication, subject to file size constraints. The total file size for the experiment presented in figure 1c is 300 Gb, for example, and there was simply no avenue for sharing files of this size when we uploaded the manuscript.

Specific comments:

1. The authors should be aware that the rates and extents of biochemical reactions are governed by the concentrations of reactants, not dimensionless relative amounts scaled to some arbitrary value such as the maximum value.

As would be expected, measured concentrations vary enormously between different cellular constituents (from ppm for K to <ppb for transition metals), with the actual values being determined by their cellular abundance (which clearly can vary) and the volume of lysis buffer employed (which is constant within a given experiment). In the introduction we remind readers of the basic cell physiology they will have learned from textbooks for intracellular concentrations of macromolecules and ions, compared with their extracellular concentrations. The primary focus of our investigation is the change in the relative abundance of cytosolic macromolecules *versus* ionic solutes over time; thus, normalisation to their maximum value (with the minimum being 0) is the clearest way for this to be communicated.

For the interested reader, the actual concentrations we measured are reported in respective supplementary figures. To further satisfy the reviewer's concern, the relevant supplementary figures (Supplementary Fig. 1d, 6b, and 12a) are now accompanied by estimation of actual cellular concentrations, subject to some simple, stated assumptions, as well as in Supplementary Table 2.

This observation is crucially important when it is reported that cellular volume is decreasing by 30 -50% over the time course of the experiments (Fig 2a, erroneously labeled 2b) while total protein content is roughly constant (Fig 1b). Thus the total intracellular protein concentration is increasing by 30 - 50% over the time course of the reaction. Since the initial total protein concentration already exceeds several hundred mg/ml, a further increase in total concentration by 30 - 50% may alter the rates and extents of biochemical reactions by as much as several orders of magnitude (Minton, chap 10 of Cellular and Molecular Physiology of Cell Volume Regulation (K Strange, ed., CRC Press, 1994; Zhou et al, Ann Rev Biophys 37: 375; 2008). These changes in absolute concentration must not be ignored.

We sincerely thank the reviewer for noticing the error in this data panel. The original (erroneously labelled) Fig. 2a misleadingly suggests a progressive decrease in overall cell volume, because there was no correction for photobleaching of the fluorescent protein reporter. In previous work with fibroblasts, consistent with the reviewer's expectation, we observed no significant change in overall cell volume, rhythmic or otherwise⁵⁶. Revised Fig. 2a now shows

the bleaching-corrected cell volume measurements, which are concordant with our previous observations and show no decrease over time. We apologise for this oversight and are very grateful to the reviewer for pointing it out – we're so used to looking for rhythms we sometimes forget to look at the trend line!

Regarding the reviewer's further point, please note that total cellular protein also includes membrane proteins, extracellular/secreted proteins, and structural proteins. Therefore, whilst cytosolic protein concentration is indeed generally found to be 200-300 mg/ml, and physiological changes within this range do affect many biological processes^{47,57}, it is not appropriate to use total protein measurements to calculate intracellular protein concentration.

2. Why is the cellular volume decreasing substantially over the time course of the experiments? Does this not indicate that the volume regulatory mechanisms of the cells being studied are seriously impaired? If so, how can the authors maintain that osmotic balance is maintained over the duration of the experiment, which is at the heart of their hypothesis?

The cell volume experiment was performed in fibroblasts, upon transient transfection with tdTomato. As described above, the apparent decrease in cell volume in the previous version of the panel was an artefact of decreasing fluorescence intensity. We corrected this in the revised panel and the results are consistent with our previous findings using a different fluorescent reporter⁵⁶ (that is, no decrease in cell volume). It is important to stress that the aim of the experiment was to explore whether measured daily rhythms in the abundance of intracellular protein, solubilised by digitonin, might be accommodated by changes in cell volume. The absence of significant circadian variation in cell volume agrees with the current paradigm of the field that cell volume is very tightly controlled, and led us to hypothesise that rhythmic compensatory ion fluxes might instead function to buffer cellular osmotic potential against changes in macromolecular solute concentration – as occurs in yeast cells¹³.

3. The authors state (top of MS p.5) that the level of macromolecular crowding in a solution is inversely proportional to the rate of diffusion of a macromolecule in that medium. I don't know what the authors mean by 'macromolecular crowding' since they never define it, and I can find no references to articles on macromolecular crowding. While the rate of macromolecular diffusion of a particular species in a crowded medium does depend upon the fraction of total volume occupied by macromolecules (one element of crowding) in the medium, it depends on many other factors as well, none of which appear to be considered in this MS. The authors' statement is highly simplistic and qualitatively inaccurate.

As far as we are aware, the first use of the term “macromolecular crowding” was by Minton *et al.*, *Biochemistry*, 1981. It is in common use, referred to by ~1000 primary research articles, with a Wikipedia entry whose introduction cites a 2001 review to state “The phenomenon of macromolecular crowding alters the properties of molecules in a solution when high concentrations of macromolecules such as proteins are present”. In preparing our original submission, and being such a common term, it simply did not occur to us that it might require explicit definition and we apologise for this oversight.

The diffusion of a given macromolecule in solution is, of course, sensitive to many factors, such as its size, temperature, pH, activity of water etc. In mammalian cytosol, under physiological conditions, the overall concentration of other macromolecules is the most relevant factor since the rest are subject to homeostatic regulation and do not change to a

significant degree over timescales of hours. Moreover, our recent proteomics analyses found no significant relationship between protein size and rhythmicity^{1,13}.

In our original submission's introduction, we stated: "Diffusion of macromolecules in aqueous solution is sensitive to the concentration of other colloidal solutes, i.e. their diffusion rate is inversely proportional to the level of macromolecular crowding". We apologise if this was misleading, but we did not claim a linear relationship, nor that diffusion of macromolecules is not affected by other factors. Macromolecular crowding is simply not the focus of this study, and so the statement was deliberately simple to aid clear communication of an auxiliary point.

However, we accept the reviewer's legitimate concern that we have oversimplified the communication of this concept. We now further substantiate the statement by including additional references to previous work e.g.^{46,48,49}. Moreover, our revised manuscript now includes additional supplementary figures demonstrating that, in aqueous solution, the apparent diffusion coefficient of quantum dots has an inverse relationship with colloidal solute concentration i.e. the extent of macromolecular crowding (Supplementary Fig. 3a). This is also true in cultured cells (volume decrease by hyperosmotic treatment and protein concentration increase following serum treatment after serum starvation – Supplementary Fig. 3b, c).

To further satisfy the reviewer's concern, in the accompanying supplementary figure legend we now discuss how change in macromolecular crowding is the most physiologically-relevant determinant of cytosolic macromolecular diffusion.

Line 1318: "The diffusion of macromolecules in solution is affected by many factors, including pH, temperature, and size, but the most physiologically-relevant determinant is macromolecular crowding."

Moreover, in our revised manuscript, we have amended the above sentence as follows:

Line 139: "Macromolecular crowding describes the concentration-dependent effect that overall macromolecular abundance has on the activities of macromolecules, as well as on the bulk solvent. Diffusion of macromolecules in the cytosol is sensitive to several factors, particularly macromolecular crowding i.e. as macromolecule concentration increases, the diffusion of macromolecules decreases^{48,49,58}."

Again, please note we do not claim a linear relationship, nor that diffusion of macromolecules is not affected by other factors, we simply focus on the element that is of direct relevance to our study.

4. The postulated inversely proportional relationship between the rate of diffusion of quantum dots in cells and total protein concentration must be examined in more detail. This is certainly not predicted theoretically nor has it been observed experimentally in model systems (Muramatsu & Minton, PNAS 85:2984; 1988). In any event, the results reported here are inconsistent with this hypothesis. We can calculate the relative total concentration of soluble protein at 24, 36, 48, and 60 hr by dividing the relative amount at these time points indicated in Fig 1b by the relative volume indicated in Fig 2a, erroneously labeled 2b (average of values from 3 curves). In the Figure reproduced below we plot the measured apparent diffusion coefficient at each of the time points against the inverse of the total protein concentration calculated as described above. These results do not support the postulated relationship between diffusion coefficient of quantum dots and soluble protein concentration. The only proper way to establish a quantitative relationship between the diffusion of quantum dots and

the composition of the intracellular medium in a medium as complex as cytoplasm is to prepare lysates of known total concentration, ion composition, and pH, to measure the rate of diffusion of quantum dots and the viscosity, and the dependence of both quantum dot diffusion rate and viscosity in the lysate upon protein concentration, ion composition, and pH. This is clearly a separate research project of its own and cannot be lumped together with reports of numerous other experiments.

We understand why the reviewer might be concerned, and again we apologise for the mistaken presentation of the uncorrected cell volume data in Fig. 2a which has led the reviewer down this line of reasoning. However, it would still not be appropriate to use the corrected total cell volumes because the total volume also includes the nucleus and other organelles, whose fractional protein content increases when the cytosolic fraction decreases (Fig. 1d). Moreover, the inverse relationship between the rate of diffusion of macromolecules in cells and macromolecular concentration is not postulated, rather it has been observed many times, with a well-established experimental and theoretical grounding^{46,48,49}. It is both intuitive and established that under physiological conditions, macromolecules pose a far greater impediment to the diffusion of other macromolecules than do smaller solutes^{48,59-61}. Finally, viscosity describes the resistance of liquids to deformation and is not relevant to our investigation, neither is cytosolic pH since this does not change over the circadian cycle (Supplementary Fig. 2).

Nonetheless, to satisfy the reviewer's legitimate concern, we further validated the expected relationship between quantum dot diffusion and macromolecule concentration (Supplementary Fig. 3a). For this we performed *in vitro* assays where we measured the apparent coefficient diffusion of QDs upon increasing physiological concentrations of macromolecules: polyethylene glycol (PEG) and bovine serum albumin (BSA). Both experiments show the expected (non-linear) inverse relationship between macromolecule concentration and coefficient diffusion. In addition, we show that the effective coefficient diffusion is not affected by changes in pH within the physiological range. We further confirm the same to be true using two different methods in cultured cells (volume decrease by hyperosmotic treatment and protein concentration increase following serum treatment after serum starvation – Supplementary Fig. 3b, c).

A more precise and quantitative description of the relationship between effective QD diffusion coefficient and cellular composition would of course be interesting, but is simply beyond the scope of this study. To remind the reviewer, we first observed daily rhythms in cytosolic protein concentration by two independent methods. Because macromolecule concentration is known to have an inverse relationship with macromolecular diffusion^{46,48,49}, we predicted the result shown in Figure 1e before performing the experiment. The utility of the quantum dot experiment is simply to test predictions suggested by observations in Figure 1b & d by a third method, i.e. the direction of change in effective diffusion over time. Macromolecular crowding is not the focus of our study and nor are our findings especially remarkable, given the established relationship between macromolecular crowding and macromolecule concentration. Our additional controls add further support to the validity of this assay, and we are not aware of any reason to doubt that the diffusion of fluorescent nanoparticles, such as quantum dots, are not sensitive to macromolecular crowding as would be expected from the published literature and as previously demonstrated by independent labs⁴⁴.

5. Expression of different proteins depends upon time in the circadian cycle. Because intracellular protein concentration is so high, changes in protein composition are likely to

result in time-dependent changes in intracellular pH, an additional modulator of intermolecular interactions in the soluble phase. pH changes would contribute to time-dependent changes in the state of association and in the solubility of individual proteins, all of which affect the osmotic pressure. The effect of pH change does not appear to have been considered in this work.

This is an important point. We have considered pH and measured it previously. By using a firefly luciferase-based sensor, we found there is no change in cytosolic pH over the circadian cycle. We now include these data in the revised version of our manuscript (Supplementary Fig. 2). In yeast cells, cytosolic pH fulfils a second messenger function⁶² as well as driving secondary active transport, whereas in mammalian cells there is no evidence to support a second messenger function and secondary active transport is mostly driven by the sodium gradient over the plasma membrane, not pH. As discussed in the revised supplementary figure legend, we can be particularly confident that there is no circadian rhythm in mammalian cytosolic pH, because we used exactly the same method to show that there is a rhythm of cytosolic pH over the yeast metabolic cycle¹³.

6. The authors' data show that the total solubility of protein is time-dependent. There is a large literature on the dependence of protein solubility upon environmental factors such as the concentration and composition of other macromolecules, osmolytes, and ions, as well as the pH. None of this literature appears to have been consulted. An understanding of these time-dependent changes depends upon understanding the interactions between the proteins and their environment.

We do not show that the total solubility of protein is time-dependent. Using two different extraction methods we show that the concentration of soluble protein extracted from the cytosolic compartment has a circadian rhythm, but that total cellular protein does not (Fig. 1b-d) and neither does cell volume (Fig. 2a). As discussed in the manuscript, this could plausibly occur through two different mechanisms which are not mutually exclusive: 1) rhythmic sequestration and liberation of protein from other cellular compartments and organelles; 2) rhythmic synthesis and degradation of cytosolic proteins. In Fig. 1, we find evidence for both and do not speculate as to whether or the other is dominant, since we previously have shown that both occur during the metabolic rhythm of yeast cells¹³. We are aware of the extensive literature on the factors that determine protein solubility, for example molecular crowding, pH and intracellular osmolyte composition⁶³⁻⁶⁶

In light of published observations, as well as our recent findings in yeast, it is most likely that a rhythm in the sequestration then liberation of cytosolic proteins into/from non-membrane bound ribonucleoprotein granules contributes to the rhythm in cytosolic soluble protein concentration we observe. We make more explicit reference to this hypothesis in the revised manuscript, as follows:

Line 133: "Whether daily cycles of cytosolic protein levels are primarily attributable to rhythmic protein synthesis/degradation or rhythmic protein sequestration/liberation warrants further investigation. However, we can exclude pH-dependent changes in cytosolic protein solubility, since no variation in cytosolic pH was detected over the cellular circadian cycle (Supplementary Fig. 2)."

This present study deals with the physiological consequences of cytosolic protein rhythms, however, rather than the relative contributions made by different underlying mechanisms that might drive them. Our investigation into the latter is in progress, and will be the basis of our next publication on the topic.

7. *The authors have not considered the very large deviations from thermodynamic ideality that exist in complex media comparable to cytosol. In the 1990's Parker and colleagues showed that mechanisms of volume regulation via compensatory ion transport were extremely sensitive to very small changes in intracellular volume and the concentration of intracellular protein. They proposed that this sensitivity was due to large changes in thermodynamic activity associated with small fractional change in intracellular protein concentration when that concentration is hundreds of mg/ml (Colclasure & Parker, J Gen Physiol 100: 1; 1992; Parker & Colclasure, Mol Cell Biochem 114:9-11; 1992; Minton et al, PNAS 89:10504; 1992). The authors should examine and discuss the relationship between this work and the present work.*

We thank the reviewer for pointing out the relevance of this older work, that anticipated contemporary understanding of the molecular mechanisms underpinning cell volume regulation. K^+ and macromolecules are well-established as the primary intracellular osmolytes in living cells, and the non-linear relationship between osmotic potential and macromolecular concentration was understood as early as 1922⁵⁷. We are happy to cite the red blood cell paper by Minton et al as requested. Indeed, in our own experiments with red blood cells, we observe similar (though much smaller) daily changes in K^+ content, without volume change, which we attribute to daily rhythms in the quaternary structure of Hb^{10,67}, which would be consistent both with our current study as well as the model by Minton *et al.* To make the importance of osmotic considerations clearer we are also now including osmotic potential measurement of PEG, BSA and KCl solutions in Supplementary Fig. 5.

References

1. Wong, D. *et al.* CRYPTOCHROME suppresses the circadian proteome and promotes protein homeostasis. *bioRxiv* 2020.05.16.099556 (2020).
2. Hoffmann, E. K., Lambert, I. H. & Pedersen, S. F. Physiology of cell volume regulation in vertebrates. *Physiol. Rev.* **89**, 193–277 (2009).
3. Gagnon, K. B. & Delpire, E. Physiology of SLC12 transporters: lessons from inherited human genetic mutations and genetically engineered mouse knockouts. *Am. J. Physiol. Physiol.* **304**, C693–C714 (2013).
4. Lauf, P. K. & Adragna, N. C. Cellular Physiology and Biochemistry K-Cl Cotransport : Properties and Molecular Mechanism. *Cell* **454435**, 341–354 (2000).
5. Arroyo, J. P., Kahle, K. T. & Gamba, G. The SLC12 family of electroneutral cation-coupled chloride cotransporters. *Mol. Aspects Med.* **34**, 288–298 (2013).
6. Gagnon, K. B. & Di Fulvio, M. A molecular analysis of the Na⁺-independent cation chloride cotransporters. *Cell. Physiol. Biochem.* **32**, 14–31 (2013).
7. Putker, M. *et al.* CRYPTOCHROMES confer robustness, not rhythmicity, to circadian timekeeping. *EMBO J.* e106745 (2021). doi:10.15252/embj.2020106745
8. Ray, S. *et al.* Circadian rhythms in the absence of the clock gene Bmal1. *Science* **367**, 800–806 (2020).
9. Lipton, J. O. *et al.* The circadian protein BMAL1 regulates translation in response to S6K1-mediated phosphorylation. *Cell* **161**, 1138–1151 (2015).
10. O’Neill, J. S. *et al.* Circadian clocks in human red blood cells. *Nature* **469**, 498–503 (2011).
11. Zambrowicz, B. P. *et al.* Wnk1 kinase deficiency lowers blood pressure in mice: a gene-trap screen to identify potential targets for therapeutic intervention. *Proc. Natl. Acad. Sci. U. S. A.* **100**, 14109–14 (2003).
12. Sengupta, S. *et al.* Regulation of OSR1 and the sodium, potassium, two chloride cotransporter by convergent signals. *Proc. Natl. Acad. Sci. U. S. A.* **110**, 18826–31 (2013).
13. O’Neill, J. S. *et al.* Eukaryotic cell biology is temporally coordinated to support the energetic demands of protein homeostasis. *Nat. Commun.* **11**, 4706 (2020).
14. Fong, C. K. *et al.* Daily rhythms in heartbeat rate are intrinsic to the zebrafish heart. *Curr. Biol.* **31**, R239–R240 (2021).
15. Uzu, T. & Kimura, G. Diuretics Shift Circadian Rhythm of Blood Pressure From Nondipper to Dipper in Essential Hypertension. *Circulation* **100**, 1635–1638 (1999).
16. Rohr, K. E. *et al.* Seasonal plasticity in GABAA signaling is necessary for restoring phase synchrony in the master circadian clock network. *Elife* **8**, (2019).
17. Sato, S. S. *et al.* Simultaneous two-photon imaging of intracellular chloride concentration and pH in mouse pyramidal neurons in vivo. *Proc. Natl. Acad. Sci. U. S. A.* **114**, E8770–E8779 (2017).
18. Akella, R. *et al.* A Phosphorylated Intermediate in the Activation of WNK Kinases. *Biochemistry* **59**, 1747–1755 (2020).
19. Wang, R., Jiang, X., Bao, P., Qin, M. & Xu, J. Circadian control of stress granules by oscillating EIF2 α . *Cell Death Dis.* **10**, 215 (2019).
20. Pattanayak, G. K. *et al.* Daily Cycles of Reversible Protein Condensation in Cyanobacteria. *Cell Rep.* **32**, 108032 (2020).
21. Hadchouel, J., Delaloy, C., Fauré, S., Achard, J.-M. & Jeunemaitre, X. Familial hyperkalemic hypertension. *J. Am. Soc. Nephrol.* **17**, 208–17 (2006).
22. Gagnon, K. B. & Delpire, E. Molecular Physiology of SPAK and OSR1: Two Ste20-Related Protein Kinases Regulating Ion Transport. *Physiol. Rev.* **92**, 1577–1617 (2012).

23. Alessi, D. R. *et al.* The WNK-SPAK/OSR1 pathway: Master regulator of cation-chloride cotransporters. *Sci. Signal.* **7**, re3–re3 (2014).
24. Sinturel, F. *et al.* Diurnal Oscillations in Liver Mass and Cell Size Accompany Ribosome Assembly Cycles. *Cell* **169**, 651–663.e14 (2017).
25. Simamora, P., Alvarez, J. M. & Yalkowsky, S. H. Solubilization of rapamycin. *Int. J. Pharm.* **213**, 25–29 (2001).
26. Huber, S. *et al.* Inhibition of the mammalian target of rapamycin impedes lymphangiogenesis. *Kidney Int.* **71**, 771–777 (2007).
27. Koumangoye, R., Bastarache, L. & Delpire, E. NKCC1: Newly Found as a Human Disease-Causing Ion Transporter. *Funct. (Oxford, England)* **2**, zqaa028 (2021).
28. Alvarez-Leefmans, F. J. (Francisco J. . & Delpire, E. *Physiology and pathology of chloride transporters and channels in the nervous system : from molecules to diseases.* (Elsevier/Academic, 2009).
29. Shekarabi, M. *et al.* WNK Kinase Signaling in Ion Homeostasis and Human Disease. *Cell Metab.* **25**, 285–299 (2017).
30. Virtanen, M. A., Uvarov, P., Hübner, C. A. & Kaila, K. NKCC1, an Elusive Molecular Target in Brain Development: Making Sense of the Existing Data. *Cells* **9**, 2607 (2020).
31. Ramanathan, C. *et al.* mTOR signaling regulates central and peripheral circadian clock function. *PLOS Genet.* **14**, e1007369 (2018).
32. Burg, M. B. & Ferraris, J. D. Intracellular organic osmolytes: Function and regulation. *J. Biol. Chem.* **283**, 7309–7313 (2008).
33. Pierce, S. K. & Warren, J. W. The taurine efflux portal used to regulate cell volume in response to hypoosmotic stress seems to be similar in many cell types: Lessons to be learned from molluscan red blood cells. *Am. Zool.* **41**, 710–720 (2001).
34. Petronini, P. G., De Angelis, E. M., Borghetti, P., Borghetti, A. F. & Wheeler, K. P. Modulation by betaine of cellular responses to osmotic stress. *Biochem. J.* **282**, 69–73 (1992).
35. Bergström, J., Fürst, P., Norée, L. O. & Vinnars, E. Intracellular free amino acid concentration in human muscle tissue. **36**, (1974).
36. Hansen, H. A. & Emborg, C. Extra- and intracellular amino acid concentrations in continuous Chinese hamster ovary cell culture. *Appl. Microbiol. Biotechnol.* **41**, 560–564 (1994).
37. Eagle, H., Piez, K. A. & Levy, M. The Intracellular Amino Acid Concentrations Required for Protein Synthesis in Cultured Human Cells. *J. Biol. Chem.* **236**, 2039–2042 (1961).
38. Xu, H. & Ren, D. Lysosomal physiology. *Annu. Rev. Physiol.* **77**, 57–80 (2015).
39. Lawrence, R. E. & Zoncu, R. The lysosome as a cellular centre for signalling, metabolism and quality control. *Nat. Cell Biol.* **21**, 133–142 (2019).
40. Brigotti, M. *et al.* Effects of osmolarity, ions and compatible osmolytes on cell-free protein synthesis. *Biochem. J* **369**, 369–374 (2003).
41. Yancey, P. H. Compatible and Counteracting Solutes: Protecting Cells from the Dead Sea to the Deep Sea. *Sci. Prog.* **87**, 1–24 (2004).
42. Garcia-Perez, A. & Burg, M. B. Role of organic osmolytes in adaptation of renal cells to high osmolality. *J. Membr. Biol.* **119**, 1–13 (1991).
43. Demetriades, C., Doumpas, N. & Teleman, A. A. Regulation of TORC1 in response to amino acid starvation via lysosomal recruitment of TSC2. *Cell* **156**, 786–99 (2014).
44. Delarue, M. *et al.* mTORC1 Controls Phase Separation and the Biophysical Properties of the Cytoplasm by Tuning Crowding. *Cell* **174**, 338–349.e20 (2018).
45. Watson, J. L., Stangherlin, A. & Derivery, E. Quantitative Chemical Delivery of

- Quantum Dots into the Cytosol of Cells. in *Quantum Dots: Applications in Biology* (eds. Fontes, A. & Santos, B. S.) 179–197 (Springer US, 2020). doi:10.1007/978-1-0716-0463-2_10
46. Ellis, R. J. J. Macromolecular crowding: obvious but underappreciated. *Trends Biochem. Sci.* **26**, 597–604 (2001).
 47. Minton, A. P., Colclasure, G. C. & Parker, J. C. Model for the role of macromolecular crowding in regulation of cellular volume. *Proc. Natl. Acad. Sci. U. S. A.* **89**, 10504–10506 (1992).
 48. Zimmerman, S. B. & Minton, A. P. Macromolecular crowding: biochemical, biophysical, and physiological consequences. *Annu. Rev. Biophys. Biomol. Struct.* **22**, 27–65 (1993).
 49. Luby-Phelps, K. Cytoarchitecture and physical properties of cytoplasm: Volume, viscosity, diffusion, intracellular surface area. *Int. Rev. Cytol.* **192**, 189–221 (1999).
 50. Kahle, K. T., Rinehart, J. & Lifton, R. P. Phosphoregulation of the Na–K–2Cl and K–Cl cotransporters by the WNK kinases. *Biochim. Biophys. Acta - Mol. Basis Dis.* **1802**, 1150–1158 (2010).
 51. Gamba, G. Molecular physiology and pathophysiology of electroneutral cation-chloride cotransporters. *Physiol. Rev.* **85**, 423–93 (2005).
 52. Delpire, E. & Gagnon, K. B. Water Homeostasis and Cell Volume Maintenance and Regulation. in *Current Topics in Membranes* **81**, 3–52 (Academic Press Inc., 2018).
 53. Okada, Y. & Maeno, E. Apoptosis, cell volume regulation and volume-regulatory chloride channels. *Comp. Biochem. Physiol. Part A Mol. Integr. Physiol.* **130**, 377–383 (2001).
 54. Stangherlin, A., Day, J. & O’Neill, J. Inductively Coupled Plasma Mass Spectrometry for Elemental Analysis in Circadian Biology. in *Methods in Molecular Biology* **2130**, 19–27 (Methods Mol Biol, 2021).
 55. Crosby, P., Hoyle, N. P. & O’Neill, J. S. Flexible measurement of bioluminescent reporters using an automated longitudinal luciferase imaging gas- and temperature-optimized recorder (ALLIGATOR). *J. Vis. Exp.* **2017**, e56623 (2017).
 56. Hoyle, N. P. *et al.* Circadian actin dynamics drive rhythmic fibroblast mobilization during wound healing. *Sci. Transl. Med.* **9**, eaal2774 (2017).
 57. Colclasure, G. C. & Parker, J. C. Cytosolic protein concentration is the primary volume signal for swelling-induced [K-Cl] cotransport in dog red cells. *J. Gen. Physiol.* **100**, 1–10 (1992).
 58. Verkman, A. S. Solute and macromolecule diffusion in cellular aqueous compartments. *Trends Biochem. Sci.* **27**, 27–33 (2002).
 59. Pink, D. A. Protein lateral movement in lipid bilayers. Stimulation studies of its dependence upon protein concentration. *Biochim. Biophys. Acta - Biomembr.* **818**, 200–204 (1985).
 60. Gaylord, N. G. & Gibbs, J. H. Physical chemistry of macromolecules. C. TANFORD. Wiley, New York, 1961. vii+710pp. \$18.00. *J. Polym. Sci.* **62**, S22–S23 (1962).
 61. Pusey, P. N. & Tough, R. J. A. Hydrodynamic interactions and diffusion in concentrated particle suspensions. *Faraday Discuss. Chem. Soc.* **76**, 123 (1983).
 62. Dechant, R. *et al.* The N-terminal domain of the V-ATPase subunit ‘a’ is regulated by pH in vitro and in vivo. *Channels (Austin)*. **5**, 4–8 (2010).
 63. Kuznetsova, I. M., Turoverov, K. K. & Uversky, V. N. What macromolecular crowding can do to a protein. *Int. J. Mol. Sci.* **15**, 23090–140 (2014).
 64. Mukherjee, S. K., Biswas, S., Rastogi, H., Dawn, A. & Chowdhury, P. K. Influence of crowding agents on the dynamics of a multidomain protein in its denatured state: a solvation approach. *Eur. Biophys. J.* **49**, 289–305 (2020).

65. Auton, M., Rösgen, J., Sinev, M., Holthauzen, L. M. F. & Bolen, D. W. Osmolyte effects on protein stability and solubility: a balancing act between backbone and side-chains. *Biophys. Chem.* **159**, 90–9 (2011).
66. Santos, J. *et al.* pH-Dependent Aggregation in Intrinsically Disordered Proteins Is Determined by Charge and Lipophilicity. *Cells* **9**, (2020).
67. Henslee, E. A. *et al.* Rhythmic potassium transport regulates the circadian clock in human red blood cells. *Nat. Commun.* **8**, 1978 (2017).

REVIEWERS' COMMENTS

Reviewer #1 (Remarks to the Author):

The authors have taken great care and consideration in revising of this manuscript, and as such it is much improved and should now be accepted for publication in Nature Comms. The authors have conducted the necessary additional experiments and these new results further support their predicted model and strengthen the paper. The authors have suitably argued their case against some additional experiments which we suggested, and we accept their rationale. The authors should be congratulated on a really fine piece of work which will no doubt have many of us in the field thinking more broadly about the implications of circadian ion transport in diverse systems.

Annie Curtis RCSI

Reviewer #2 (Remarks to the Author):

The authors have satisfactorily addressed all my concerns and I have no further criticisms.

Reviewer #3 (Remarks to the Author):

The authors have addressed my comments in an appropriate manner

Reviewer #4 (Remarks to the Author):

The authors have answered all of my comments and questions.

Reviewer #5 (Remarks to the Author):

For the most part, the authors have made suitable revisions to accommodate my questions and comments on the original submission. Most importantly, they have corrected the figure that implied a large time-dependent decrease in cell volume over the time course of their experiments. They no longer erroneously state that the diffusion coefficient of quantum dots is inversely proportional to the total concentration of soluble protein. However, their statement that the rate of diffusion decreases as the protein concentration in cytosol increases is not quite correct. The rate of diffusion decreases as the total concentration of all soluble macromolecules, not just proteins, increases. They should justify neglect of possible changes in nucleic acid or polysaccharide concentration, either by showing that the total concentrations of the other macromolecular species are negligible relative to that of proteins, or by showing that their concentrations do not change with time.

Minor comment: In the legend to supplementary Figure 5, it is stated that the osmotic pressure of PEG and hemoglobin show a nonlinear dependence on concentration due to the "increasing proportions of solvent molecules being affected by the extended hydration shells around macromolecules". This is incorrect. The nonlinear dependence of the osmotic pressure of hemoglobin has been shown to be accurately accounted for over a very wide range of concentration by the assumption that Hb molecules behave as hard spherical particles with a size very close to that of a hemoglobin molecule, and that hydration does not play a significant role (Ross & Minton, J Mol Biol 112: 437; 1977).

Reviewer: *italic*
Authors: plain text

REVIEWER COMMENTS

Reviewer #1 (Remarks to the Author):

The authors have taken great care and consideration in revising of this manuscript, and as such it is much improved and should now be accepted for publication in Nature Comms. The authors have conducted the necessary additional experiments and these new results further support their predicted model and strengthen the paper. The authors have suitably argued their case against some additional experiments which we suggested, and we accept their rationale. The authors should be congratulated on a really fine piece of work which will no doubt have many of us in the field thinking more broadly about the implications of circadian ion transport in diverse systems.

Annie Curtis RCSI

Thank you.

Reviewer #2 (Remarks to the Author):

The authors have satisfactorily addressed all my concerns and I have no further criticisms.

Thank you.

Reviewer #3 (Remarks to the Author):

The authors have addressed my comments in an appropriate matter

Thank you.

Reviewer #4 (Remarks to the Author):

The authors have answered all of my comments and questions.

Thank you.

Reviewer #5 (Remarks to the Author):

For the most part, the authors have made suitable revisions to accommodate my questions and comments on the original submission. Most importantly, they have corrected the figure that implied a large time-dependent decrease in cell volume over the time course of their experiments. They no longer erroneously state that the diffusion coefficient of quantum dots is inversely proportional to the total concentration of soluble protein. However, their statement that the rate of diffusion decreases as the protein concentration in cytosol increases is not quite correct. The rate of diffusion decreases as the total concentration of all soluble macromolecules, not just proteins, increases. They should justify neglect of possible changes in nucleic acid or polysaccharide concentration, either by showing that the total concentrations of the other macromolecular species are negligible relative to that of proteins, or by showing that their concentrations do not change with time.

Proteins are the most abundant soluble macromolecules in mammalian cells by an order of magnitude¹⁻³. Consistent with this we find protein concentrations in our cytosolic extracts are

consistently >10-fold higher than RNA, the second most abundant cytosolic macromolecule. Thus whilst we do not exclude the contribution of other classes of macromolecules to the rhythm in macromolecular crowding we have observed, we have understandably chosen to focus on that which we believe to be the class of greatest relevance to the phenomenon under investigation. To more clearly communicate this we have made the following addition to the text:

Page 5, Line 145. “Proteins are the most abundant macromolecule in mammalian cells¹. Therefore, to further validate the observed rhythm in cytosolic protein concentration, we measured the diffusion of QDs in the cytosol⁵¹, reasoning that changes in cytosolic protein concentration during the circadian cycle should inversely impact the diffusion of QDs.”

Minor comment: In the legend to supplementary Figure 5, it is stated that the osmotic pressure of PEG and hemoglobin show a nonlinear dependence on concentration due to the “increasing proportions of solvent molecules being affected by the extended hydration shells around macromolecules”. This is incorrect. The nonlinear dependence of the osmotic pressure of hemoglobin has been shown to be accurately accounted for over a very wide range of concentration by the assumption that Hb molecules behave as hard spherical particles with a size very close to that of a hemoglobin molecule, and that hydration does not play a significant role (Ross & Minton, J Mol Biol 112: 437; 1977).

We thank the referee for referring us to this interesting theoretical work. The reviewer will agree that any model is simply a quantitative form of scientific hypothesis. That any given model might reproduce empirical observation, does not mean it explains the basis for a given natural phenomenon. Newton’s model of gravitation successfully accounted for empirical observation, for example, until superseded Einstein’s. Given that Minton's model is not widely accepted in the field (3 citations in 50 years), and that there is broad consensus that the % of solvent molecules in hydration shells (otherwise known as cybotactic water) varies non-linearly as macromolecular concentration increases⁴⁻⁹, we believe that the figure legend’s summary of the current model is uncontroversial. To assuage the reviewer’s doubts, however, we have made the following addition to the supplementary figure legend:

Page 52, Line 1383. “We note that other interpretations for this phenomenon have been proposed (Ross & Minton, J Mol Biol 112: 437; 1977).”

REFERENCES

1. Feijó Delgado, F. *et al.* Intracellular Water Exchange for Measuring the Dry Mass, Water Mass and Changes in Chemical Composition of Living Cells. *PLoS One* **8**, e67590 (2013).
2. Albayrak, C. *et al.* Digital Quantification of Proteins and mRNA in Single Mammalian Cells. *Mol. Cell* **61**, 914–924 (2016).
3. Hudder, A., Nathanson, L. & Deutscher, M. P. Organization of mammalian cytoplasm. *Mol. Cell. Biol.* **23**, 9318–26 (2003).
4. Cameron, I. L. & Fullerton, G. D. A model to explain the osmotic pressure behavior of hemoglobin and serum albumin. *Biochem. Cell Biol.* **68**, 894–898 (1990).
5. Kanal, K. M., Fullerton, G. D. & Cameron, I. L. A study of the molecular sources of nonideal osmotic pressure of bovine serum albumin solutions as a function of pH.

- Biophys. J.* **66**, 153–160 (1994).
6. Fullerton, G. D., Kanal, K. M. & Cameron, I. L. Osmotically unresponsive water fraction on proteins: Non-ideal osmotic pressure of bovine serum albumin as a function of pH and salt concentration. *Cell Biol. Int.* **30**, 86–92 (2006).
 7. Ebbinghaus, S. *et al.* An extended dynamical hydration shell around proteins. *Proc. Natl. Acad. Sci. U. S. A.* **104**, 20749–20752 (2007).
 8. Persson, E. & Halle, B. *Cell water dynamics on multiple time scales. PNAS April 29*, (2008).
 9. Aggarwal, L. & Biswas, P. Hydration Water Distribution around Intrinsically Disordered Proteins. *J. Phys. Chem. B* **122**, 4206–4218 (2018).